# Maximum Total Correlation Reinforcement Learning

**Bang You**[1][2]  **Puze Liu**[3]  **Huaping Liu**[2]  **Jan Peters**[3][4][5][6]  **Oleg Arenz**[3]

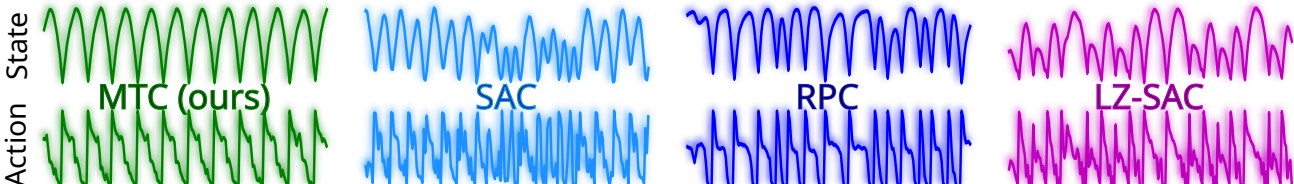

*Figure 1.* Maximizing the total correlation within trajectories results in more consistent behavior. As shown in our experiments, this consistency increases robustness to noise and dynamics changes.

## Abstract

Simplicity is a powerful inductive bias. In reinforcement learning, regularization is used for simpler policies, data augmentation for simpler representations, and sparse reward functions for simpler objectives, all that, with the underlying motivation to increase generalizability and robustness by focusing on the essentials. Supplementary to these techniques, we investigate how to promote simple *behavior* throughout the episode. To that end, we introduce a modification of the reinforcement learning problem that additionally maximizes the total correlation within the induced trajectories. We propose a practical algorithm that optimizes all models, including policy and state representation, based on a lower-bound approximation. In simulated robot environments, our method naturally generates policies that induce periodic and compressible trajectories, and that exhibit superior robustness to noise and changes in dynamics compared to baseline methods, while also improving performance in the original tasks.

## 1. Introduction

Reinforcement learning (RL) is currently the preferred approach for many challenging, practical control problems, as it can learn complex neural network policies that effectively tackle the given task. For example, in robotics, reinforcement learning is widely used to learn visuomotor policies for quadrupedal and bipedal locomotion (Lee et al., 2020a; Radosavovic et al., 2024). However, since RL is a learning-based method, it is prone to picking up spurious correlations between high-dimensional sensory inputs and desired actions, which can lead to brittle policies, that fail under slight, natural variations in the state. An important countermeasure involves training policies using domain randomization, in particular in the sim-to-real setting, where the policy is learned in several varying simulation environments. Yet, even in such data-intense settings, it remains unclear whether we can obtain a sufficiently diverse training distribution to learn policies that transfer to more complex real-world scenarios, such as those involving robot controllers that need to interact with humans.

Consequently, there is a growing interest in exploring additional techniques that add inductive biases to obtain simpler, less brittle policies, for example, by limiting the amount of state information used by the policy (Goyal et al., 2018; Igl et al., 2019; Lu et al., 2020), or by maximizing the predictive information within learned representations (Lee et al., 2020b). Such information-theoretic biases have already been extended to sequences, to account for the sequential nature of reinforcement learning. Namely, RPC (Eysenbach et al., 2021) aims to learn better representations by limiting the information between state-sequences and embedding-sequences, and LZ-SAC (Saanum et al., 2023) improves the predictability of the next action given the history of actions. However, these formulations only focus on specific

---

[1]School of Information Engineering, Wuhan University of Technology, Wuhan, China [2]Department of Computer Science, Tsinghua University, Beijing, China [3]Intelligent Autonomous Systems Lab, Technische Universität Darmstadt, Darmstadt, Germany [4]Deutsches Forschungszentrum für Künstliche Intelligenz (DFKI), Germany [5]Hessian Centre for Artificial Intelligence (Hessian.AI) [6]Centre for Cognitive Science (CogSci). Correspondence to: Huaping Liu <hpliu@tsinghua.edu.cn>.

aspects of the behavior—either state-consistency or action-consistency—without considering the complete behavior.

Instead, we propose a novel inductive bias that operates on the level of trajectories. Specifically, we aim to learn policies that produce simple, consistent, and therefore compressible trajectories, exhibiting a tendency towards open-loop behavior. We hypothesize that such behavior is not only more interpretable and predictable, but also more robust to slight variations in the state that the agent inevitably encounters due to sensor noise or unmodeled dynamic effects. Following the spirit of Newton's first rule (Newton et al., 1934, Book III), we are to admit no more reactions to the state variations than such as are both relevant and sufficient to achieve high expected return. Intuitively, we expect a given behavior that performed well for previous variations to also perform well for future variations. We introduce this inductive bias by means of the additional objective of maximizing the total correlation within the generated trajectories. This total correlation corresponds to the amount of information that we can save by using a joint encoding of all states and actions within trajectories, compared to compressing all time steps independently. By maximizing total correlation, the agent is encouraged to produce compressible and predictable trajectories, and thereby biased towards open-loop behavior such as clean periodic gaits, without preventing it from performing adaptations when necessary.

The main contributions of our work are as follows. We introduce the maximum total correlation reinforcement learning problem (MTC-RL), which extends the typical RL formulation with an additional objective of maximizing trajectory total correlation. We derive a lower-bound approximation of the total correlation and use it to propose a practical algorithm for MTC-RL, based on soft-actor critic (Haarnoja et al., 2018). Our algorithm features an adaptation scheme to automatically adapt the coefficient of the total correlation objective by treating it as the Lagrangian multiplier of a constrained optimization problem. We empirically evaluate our algorithm on simulated robotic control tasks and show that the learned policies induce more periodic and better compressible trajectories than baseline methods (Eysenbach et al., 2021; Saanum et al., 2023), leading to an improve in performance, as well as robustness to observation noise, action noise, and changes in the system dynamics.[1]

## 2. Related Work

Information theory provides effective tools to solve problems in RL (Peters et al., 2010; Memmel et al., 2022; Tishby & Zaslavsky, 2015; Ma et al., 2023; Chakraborty et al., 2023), such as representation learning (Oord et al., 2018;

Gao et al., 2024), and generalization (Goyal et al., 2018). Motivated by the InfoMax principle (Bell & Sejnowski, 1995), some previous RL methods preserve mutual information to extract useful representations from observations, and have achieved improvement in terms of performance and robustness on downstream tasks (Kim et al., 2019; Laskin et al., 2020; Mazoure et al., 2020; Rakelly et al., 2021; Dunion et al., 2024). These methods usually maximize mutual information in single transitions. In contrast, our approach maximizes the total interdependencies within the trajectories of an agent. Moreover, instead of using separate objectives for policy and state encoder, we use a unified objective to optimize policy and representations with respect to the consistency within the resulting trajectories.

Total correlation is a fundamental concept in information theory to qualify the statistical dependency among multiple random variables (Watanabe, 1960). Previous methods have shown that total correlation is an effective tool to enhance machine learning models in many tasks, such as disentangled representation learning (Steeg, 2017; Gao et al., 2019) or structure discovery (Ver Steeg & Galstyan, 2014). Our work extends these results to the RL setting by observing that the agent can actively change its behavior to maximize consistency within state and action sequences. Our method is also related to previous methods that endow RL agents with robust behavior (Tessler et al., 2019; Tanabe et al., 2022; Reddi et al., 2023; Shi & Chi, 2024). While these methods have proposed purpose-designed methods to achieve robustness benefits, we focus on demonstrating that maximizing the total correlation is a simple and effective task-independent solution for improving robustness.

The principle of simplicity has garnered substantial attention in constructing learning agents (Chater & Vitányi, 2003; Tishby & Zaslavsky, 2015; Grau-Moya et al., 2018; Igl et al., 2019; Goyal et al., 2018; Tishby & Polani, 2010; Leibfried & Grau-Moya, 2020). Some recent works induce simple policies by imposing temporal consistency in actions. For example, Saanum et al. (2023) propose to capture the temporal consistency in action sequences and induce simple behaviors by incorporating the preference for consistent actions into the reward function. Another class of methods enforces temporal consistency in latent representations of states to obtain policies that produce simple behaviors. For instance, RPC (Eysenbach et al., 2021) learns policies that visit states whose representations are temporally consistent in individual transitions, by minimizing the mutual information between a sequence of observations and a sequence of their representations. In contrast, our total correlation objective maximizes the consistency among sequences of state representations and actions. This difference, which corresponds to learning dynamic models that predict the future from a history of actions and states, allows the agent to achieve consistent behavior throughout whole trajectories.

---

[1]Our code is publicly available at `https://github.com/BangYou01/MTC`.

Our approach is also related to previous approaches that extract temporally consistent representations from observations by learning latent dynamics models (Guo et al., 2022; Hansen et al., 2022). Unlike these approaches that only consider temporal consistency in representations of states, our total correlation objective aims to maximize the consistency among the whole trajectories of state representations and actions. As shown in our experiments (Fig. 5), additionally enforcing the consistency within action sequences by learning the action prediction model improves the performance in the presence of environmental perturbations. Some approaches learn representations that discard unnecessary information in raw states via the Fourier transform (Li & Pathak, 2021; Ye et al., 2023). In contrast, our approach filters out irrelevant state information by maximizing total correlation within trajectories of representations and actions.

## 3. Preliminaries and Notations

In this section, we provide a brief overview of background for information theory reinforcement learning, and introduce the notation used throughout the paper.

### 3.1. Information Theory Background

Mutual information (MI) is a commonly used statistical dependency measurement in machine learning (Alemi et al., 2017). Given two random variables $x_1$ and $x_2$, their mutual information is defined as:

$$\mathcal{I}(x_1; x_2) = \mathbb{E}_{x_1, x_2} \left[ \log \frac{p(x_1, x_2)}{p(x_1)p(x_2)} \right].$$

Total correlation, or multi-information, generalizes mutual information to more than two random variables (Watanabe, 1960; Studenỳ & Vejnarová, 1998). The total correlation $\mathcal{C}(x_1; x_2; \ldots; x_n)$ of $n$ random variables $x_i$, is defined as the Kullback-Leibler (KL) divergence between the joint distribution and the product of their marginals,

$$\mathcal{C}(x_1; x_2; \ldots; x_n) = \mathbb{E}_{x_1, x_2, \ldots, x_n} \left[ \log \frac{p(x_1, x_2, \ldots, x_n)}{\prod_{i=1}^{n} p(x_i)} \right].$$

This KL divergence corresponds to the expected amount of information (measured in nats), that we can save when transmitting the sequence $(x_1, \ldots, x_n)$ using a code that is optimized with respect to the complete sequence, compared to independently encoding each random variable $x_i$.

### 3.2. Markov Decision Process

We formulate the maximum total correlation reinforcement learning problem in a finite horizon Markov decision process (MDP), denoted by the tuple $\mathcal{M} = (\mathcal{S}, \mathcal{A}, p, r, T)$, where $\mathcal{S}$ is the state space, $\mathcal{A}$ is the action space, $p(s_{t+1}|s_t, a_t)$ is the stochastic dynamic model, $r(s, a)$ is the reward function,

and $T$ is the time horizon. At each time step, the agent observes the current state $s_t$ and selects its actions $a_t$ based on its stochastic policy $\pi(a_t|s_t)$ and then receives the reward $r(s_t, a_t)$. The original reinforcement learning objective is to maximize the expected cumulative rewards $\mathbb{E}_\tau \left[ \sum_{t=1}^{T} r_t \right]$ where $\tau = (s_1, a_1, s_2, a_2, \ldots)$ denotes the agent's trajectory. As typically not all state information is relevant for choosing the optimal action, we will assume, without loss of generality, that the policy chooses the action based on a latent variable $z_t \sim f(z_t|s_t)$ using a learned encoder $f$. We refer to the parameters of encoder and policy by $\theta$ and $\phi$, respectively, and we sometimes write $\pi_\phi$ and $f_\theta$ to make this dependency explicit, however, we typically omit the subscript for brevity.

While we use the finite horizon setting for formulating MTC-RL to ensure that the total correlation of trajectories takes finite values, we will transition to the infinite horizon setting in Section 4.3, by letting $T$ go to infinity and introducing a discount factor $\gamma$. In the infinite horizon setting, which underlies the practical implementation used in our experiments, the agent aims to maximize the expected discounted cumulative rewards $\mathbb{E}_\tau \left[ \sum_{t=1}^{\infty} \gamma^t r_t \right]$.

## 4. Maximum Total Correlation Reinforcement Learning

In this section, we introduce the maximum total correlation reinforcement learning problem, derive a variational lower bound on the total correlation, and use it to formulate an optimization problem that can be solved with existing reinforcement learning methods.

### 4.1. Problem Formulation and Motivation

We want to bias the policy towards producing simpler behavior in order to increase its robustness towards state-, action- or dynamics-perturbations. We quantify the simplicity of the behavior by the total correlation of the complete trajectories induced by the policy, which corresponds to their compressibility in an information-theoretic sense. More specifically, we extend the vanilla reinforcement learning objective by introducing the additional objective of maximizing the total correlation within the trajectory of latent state representations and actions,

$$\max_{\theta, \phi} \quad \mathbb{E}_{\pi_\phi, f_\theta} \left[ \left[ \sum_{t=1}^{T} r(s_t, a_t) \right] + \alpha \mathcal{C}(z_1; a_1; \ldots; a_{T-1}; z_T) \right].$$
(1)

where the hyper-parameter $\alpha$ controls the trade-off between both objectives.

Using the latent representation $z$ rather than the raw states $s$

for the total correlation objective serves two main purposes. Firstly, by restricting our total correlation objective to task-relevant state information, we focus on learning behavior that is consistent only with respect to aspects of the state that actually matter for the task. The second motivation for formulating the total correlation with respect to the learned state representation is to not only learn more consistent behavior, but also more consistent representations $z$. By penalizing unnecessary variations in the representation, we aim to learn representations that are more robust to irrelevant variations in the state.

## 4.2. A Variational Bound on Total Correlation

The total correlation objective in Eq. 1 can not be decomposed into a sum of step-rewards and involves probability distributions that are typically not available in analytic form. Hence, we replace it with a variational lower bound, using a parameterized history-based latent dynamics model $q_\eta(z_{t+1}|z_{1:t}, a_{1:t})$ and a parameterized history-based action prediction model $q_\chi(a_t|z_{1:t}, a_{1:t-1})$,

$$\mathcal{C}(z_1; a_1; \ldots; a_{T-1}; z_T) \geq \widetilde{\mathcal{C}}(z_1; a_1; \ldots; a_{T-1}; z_T)$$
$$= \mathbb{E}_{\pi,f}\left[\sum_{t=1}^{T-1}\left[\log \frac{q_\eta(z_{t+1}|z_{1:t}, a_{1:t})q_\chi(a_t|z_{1:t}, a_{1:t-1})}{f_\theta(z_{t+1}|s_{t+1})\pi_\phi(a_t|s_t)}\right]\right].$$
(2)

Please refer to Appendix A.1 for the derivation. The contribution of a given time step $t$ to the lower bound is large when the next latent state and the next action can be well predicted based on the history, while accounting for the irreducible uncertainty due to the stochastic encoder $f$ and the policy $\pi$. Hence, this mechanism encourages consistent trajectories. As shown in our experiments, both state consistency and action consistency are significantly improved when using the lower bound $\widetilde{\mathcal{C}}$ within the MTC-RL objective (see Figure 1), which demonstrates that the lower bound captures important aspects of the total correlation.

## 4.3. A Tractable Optimization Problem

By plugging the lower bound $\widetilde{\mathcal{C}}$ in Eq. 2 into the objective function Eq. 1, we obtain the tractable objective

$$\max_{\theta,\phi,\eta,\chi} \quad \mathbb{E}_{\pi_\phi,f_\theta}\left[r(s_T, a_T) + \sum_{t=1}^{T-1}\left[r(s_t, a_t)\right.\right.$$
$$\left.\left.+ \alpha\left[\log \frac{q_\eta(z_{t+1}|z_{1:t}, a_{1:t})}{f_\theta(z_{t+1}|s_{t+1})} + \log \frac{q_\chi(a_t|z_{1:t}, a_{1:t-1})}{\pi_\phi(a_t|s_t)}\right]\right]\right]$$
(3)

that we optimize with respect to the parameters of the policy, encoder, and latent dynamics model.

The policy is, thus, optimized with respect to the information-regularized reward function

$$r^*(s_t, a_t, s_{t+1}) = r(s_t, a_t, s_{t+1})$$
$$+ \alpha\left(\log \frac{q_\eta(z_{t+1}|z_{1:t}, a_{1:t})q_\chi(a_t|z_{1:t}, a_{1:t-1})}{f_\theta(z_{t+1}|s_{t+1})\pi_\phi(a_t|s_t)}\right).$$
(4)

The modified reward biases the policy towards states for which the latent representation can be well predicted based on the history, relative to the uncertainty in the encoder predictions, and towards actions that can be well predicted by the action prediction model, relative to the uncertainty of the policy. Although simplified in notation, this reward function also depends on the history and on the current parameters of the policy and encoder. The dependence on past states might suggest the need for a history-based policy; however, our ablations show that providing only the current state as input to the policy can achieve similar performance. Furthermore, despite the non-stationarity introduced by the reward function's dependence on policy and encoder parameters, we did not observe learning instabilities.

The latent history-based dynamics and action prediction models get trained using maximum likelihood, and the encoder and policy get biased towards the history-based predictions, due to the additional objectives of minimizing the KL divergence towards history-based models.

For the practical implementation, we switch to the infinite horizon problem setting by letting $T \to \infty$, and introducing the discount factor $\gamma$, that is, we optimize the final objective

$$\max_{\theta,\phi,\eta,\chi} \quad \mathbb{E}_{\pi_\phi,f_\theta}\left[\sum_{t=1}^{\infty}\gamma^t r^*(s_t, a_t, s_{t+1})\right].$$
(5)

As clarified in Appendix A.4 this objective corresponds to maximizing a lower bound on a natural extension of total correlation to infinite sequences.

## 4.4. Maximum Total Correlation Soft Actor Critic

Our total correlation regularized reinforcement learning problem in Eq. 5 can be optimized straightforwardly with existing RL methods. For our experiments we implement MTC on top of soft actor-critic (SAC) (Haarnoja et al., 2018). As an actor-critic method, SAC alternates between estimating the Q function (policy evaluation) and improving the policy with respect to the Q function (policy improvement). SAC considers the maximum entropy RL setting, that is, it has the additional objective of maximizing the entropy of the policy, and therefore, it computes the soft-Q function $Q_{\text{soft}}^\pi(s, a)$ during policy evaluation, which also accounts for the expected future entropy of the policy. For our policy evaluation, we do not need to make any modifications to SAC, besides replacing the original reward function $r(s_t, a_t)$ with the regularized reward $r^*(s_t, a_t, s_{t+1})$. Hence, we also learn the soft-Q function and use common techniques such

as target nets (Mnih et al., 2015) and dual Q nets (Fujimoto et al., 2018; Haarnoja et al., 2018).

For policy improvement, however, we also optimize the dynamics model, the action prediction model and the encoder along with policy. While the prediction models are, thus, trained on the replay buffer instead of using on-policy samples, which slightly deviates from the derived update and may increase the gap of our lower bound, this change allows for an easy integration of the total-correlation regularizer for off-policy optimization. Similar to RPC (Eysenbach et al., 2021), we express the soft-Q function in terms of the regularized reward and the soft Q-function of the next time step, to arrive at the following objective,

$$
\begin{aligned}
\max_{\theta,\phi,\eta,\chi} \quad & \mathbb{E}_{\mathcal{D},\pi_\phi,f_\theta}\Big[ - (\alpha + \beta) \log(\pi_\phi(a_t|s_t)) \\
& + \alpha\Big( \log \frac{q_\eta(z_{t+1}|z_{1:t}, a_{1:t})q_\chi(a_t|z_{1:t}, a_{1:t-1})}{f_\theta(z_{t+1}|s_{t+1})} \Big) \\
& + \gamma\Big( Q_{\text{soft}}^\pi(s_{t+1}, a_{t+1}) - (\alpha + \beta) \log(\pi_\phi(a_{t+1}|s_{t+1})) \Big)\Big],
\end{aligned}
$$
(6)

where $s_{1:t+1}$ and $a_{1:t}$ are sampled from the replay buffer $\mathcal{D}$, $a_{t+1}$ is sampled from the current policy, and all embeddings $z_{1:t+1}$ are sampled from the current encoder. The coefficient $\beta$ corresponds to the weight of the entropy regularizer of SAC.

Furthermore, instead of choosing the hyperparameter $\alpha$ directly, we optimize it with respect to a desired bound $I_p$, by minimizing the dual objective

$$
L(\alpha) = \alpha\Big( \log \frac{q_\eta(z_{t+1}|z_{1:t}, a_{1:t})q_\chi(a_t|z_{1:t}, a_{1:t-1})}{f_\theta(z_{t+1}|s_{t+1})\pi_\phi(a_t|s_t)} - I_p\Big).
$$
(7)

## 5. Experimental Evaluation

We performed experiments to investigate how our total correlation objective compares to vanilla soft-actor critic (Haarnoja et al., 2018) and the closely related alternative methods RPC (Eysenbach et al., 2021), LZ-SAC (Saanum et al., 2023) and SPAC (Saanum et al., 2023) in terms of performance on the original RL objective (Sec. 5.1 and Sec. 5.4), robustness to noise, dynamics mismatch and spurious correlation (Sec. 5.2), and consistency of the resulting trajectories (Sec. 5.3). Furthermore, we performed ablations to investigate the effects for components of our model and total correlation constraints $I_p$ (Sec. 5.5).

We build MTC on top of the open source implementation of SAC by Yarats et al. (2021). Whereas the official implementation of LZ-SAC provided by Saanum et al. (2023) also uses this SAC implementation, the original implementation of RPC provided by Eysenbach et al. (2021) is based on the SAC implementation from TF-Agents. To ensure a reliable

and fair comparison to RPC, we compare MTC to RPC implemented by its original code (referred to as RPC-Orig in Table. 1) and to our implementation of RPC built on top of the same SAC codebase as MTC and LZ-SAC (referred to as RPC). Please refer to Appendix B for details on the implementations of the different approaches.

### 5.1. Performance

In our first set of experiments we evaluate the performance on the original reinforcement learning problem. Table. 1 shows the final performance of our method and baselines on eight continuous control tasks from the DeepMind Control (DMC) (Tassa et al., 2018), a commonly used open-source simulated benchmark in RL settings. Learning curves are shown in Fig. 7 in Appendix C.1. MTC achieves better average asymptotic performance than baselines on the majority of the tasks. In particular, MTC outperforms SAC on five tasks, Hopper Stand, Finger Spin, Cheetah Run, Walker Run, and Quadruped Walk. These results suggest that inducing simple policies by maximizing the total correlation also benefits policy learning.

### 5.2. Zero-shot Robustness

Our main motivation for learning consistent behavior and representations is to improve robustness by focusing on the essentials. Our policies are biased to produce trajectories that have fewer variations, so we expect that they are more robust to unseen disturbances. Hence, we evaluated our method and baselines in terms of zero-shot robustness to observation, action , and dynamics perturbations.

**Robustness to observation perturbations.** We first investigate how observation perturbations affect policy performance by injecting Gaussian noise into the observations, $s_t \leftarrow s_t + \epsilon$, where noise $\epsilon$ is sampled from a Gaussian distribution, $\epsilon \sim \mathcal{N}(0, \text{diag}(\sigma^2))$ with standard deviation $\sigma$. Using the same tasks as before, we evaluate our method and baselines on a series of noise strength $\sigma \in [0.02, 0.04, 0.06, 0.08, 0.1]$. To compare the robustness across all eight tasks, we normalized the scores by the score achieved by the best method on each task. The aggregated robustness to observation perturbations with different noise strengths is shown for different methods in Figure 2 (left). MTC achieves the best aggregated performance when observations are perturbed with Gaussian noise.

**Robustness to action perturbations.** As our total correlation objective encourages consistent actions, we also expect an improvement in terms of robustness to action perturbations. Hence, we add Gaussian noise to the actions, $a_t \leftarrow a_t + \epsilon$, where $\epsilon \sim \mathcal{N}(0, \text{diag}(\sigma^2))$ with noise strength $\sigma$. However, please note that we had to clip the values of the noisy actions to be within $[-1, 1]$, due to requirements of the simulator.

*Table 1.* Scores (means over 20 seeds with 90% confidence interval) achieved by our method and baselines on eight DMC tasks at 1 million environment steps. MTC achieves better or at least comparable asymptotic performance than all baselines. In particular, MTC outperforms LZ-SAC, RPC, and SPAC by a large margin on five tasks.

| Scores | MTC | RPC | RPC-Orig | LZ-SAC | SPAC | SAC |
|---|---|---|---|---|---|---|
| Acrobot Swingup | **184 ± 24** | **132 ± 31** | 20±3 | 100±22 | 110±29 | **154 ± 29** |
| Hopper Stand | **933± 12** | 568 ± 96 | 476± 101 | 593± 88 | 213±69 | 683± 114 |
| Finger Spin | **985± 2** | 869 ± 19 | 921 ± 13 | 805± 38 | 136±121 | 955± 18 |
| Walker Walk | **967± 2** | 940± 21 | 951 ± 2 | 939± 26 | 883±76 | **962± 7** |
| Cheetah Run | **874± 21** | 772± 57 | 636 ± 10 | 787± 17 | 458±52 | 811± 36 |
| Quadruped Walk | **944± 5** | 842 ± 77 | 598 ± 108 | 595 ± 110 | 505±185 | 738± 93 |
| Walker Run | **790 ± 9** | **778 ± 25** | 604± 29 | 732± 22 | 347±95 | 767± 13 |
| Walker Stand | **983± 2** | **980 ± 5** | 971 ±1 | 977± 2 | 931±38 | **985± 2** |

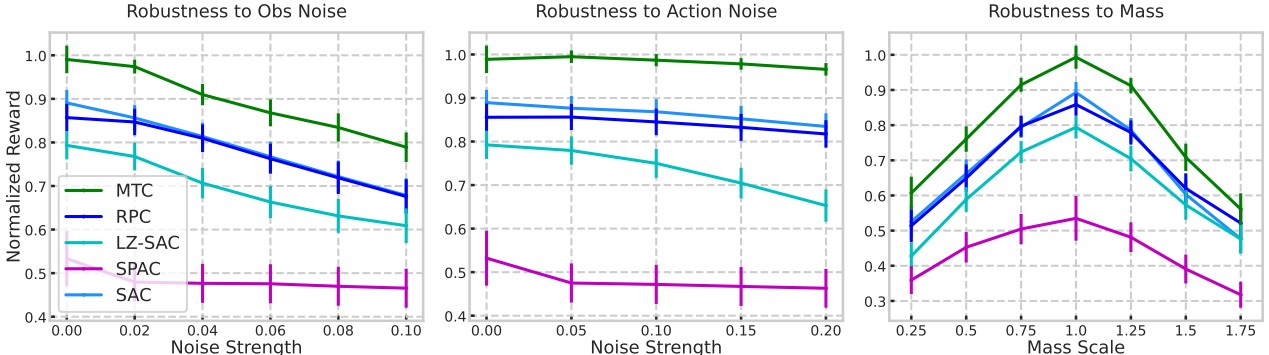

*Figure 2.* We evaluated the robustness towards observation noise (left), action noise (middle) and mass changes (right) on eight tasks from DMC benchmarks. The plots show the normalized mean rewards averaged over 20 independent runs and 8 tasks, with error bars representing 90% confidence interval. For each task we normalized the return by the mean return of the best method. Each run includes 30 evaluation trajectories. MTC achieves better aggregated performance than baselines in the presence of perturbations to observations and actions, while also obtaining higher mean rewards when the body mass is changed slightly.

Overall, MTC achieves higher average rewards than all baselines even in the presence of strong action perturbations, see Fig. 2 (middle). Notably, our approach outperforms SAC in robustness to action perturbations with different noise strengths, indicating that maximizing trajectory total correlation improves robustness to action perturbations.

**Robustness to dynamics mismatches.** We also expect simple behavior to be more robust towards deviations between the dynamics encountered during testing compared to the dynamics used for training. We test the effects of dynamics mismatch by scaling the mass of each robot body during evaluation. We evaluate six different scaling factors in each environment, namely $[0.25, 0.5, 0.75, 1.25, 1.5, 1.75]$, and present the aggregated results on eight tasks in Fig. 2 (right). Overall, MTC obtains higher averaged scores than all baselines in the presence of small dynamics changes.

**Robustness to spurious correlations.** We further performed experiments to evaluate the robustness of MTC to spurious correlations on the Walker Stand task. To that end,

we introduced additional state dimensions that are not controllable by the actor, but instead follow a fixed Gaussian transition model. These distractors are not correlated with the remaining states, the actions nor the reward that the agent receives, although by coincidence, it might appear that such correlations exist, resulting in spurious correlations. Fig. 11 in Appendix C.9 shows the performance of our method, RPC and SAC. The plot shows the mean over 10 seeds, with a 90% confidence interval. MTC significantly outperforms RPC and SAC in rewards, suggesting that MTC improves robustness to spurious correlations.

### 5.3. Trajectory Consistency

Arguably, the consistency of a behavior can be most straightforwardly judged by visualizing it. Hence, we generated plots of the state and action trajectories for MTC and all baselines on the Finger Spin task. We already showed the trajectories for the first joint in Fig. 1. The remaining joints are shown in Figure 12 and Figure 13 in Appendix C.10.

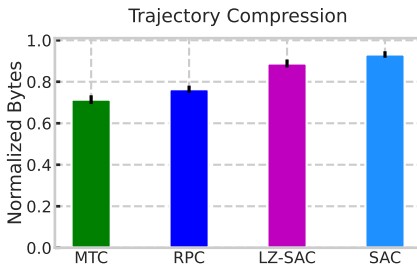

*Figure 3.* The compressed state-action trajectories obtained by MTC have smallest file size in expectation.

Based on these visualizations, we argue that MTC produces the simplest and most consistent trajectories, characterized by highly cyclical patterns.

To support this qualitative assessment, inspired by Saanum et al. (2023), we use lossless compression algorithms to quantify the compressibility of trajectories produced by learned policies. We round the collected state-action trajectories to one digit behind the decimal point, save them as .npy-files and compress them using bzip2. Rounding the floating point numbers was necessary to achieve meaningful results because otherwise the highly random insignificant bits would dominate, leading to high variance in the resulting file sizes. Figure 3 shows the normalized average file sizes in bytes among 30 trajectories of 1000 steps for each of the 8 tasks, with error bars representing 90% confidence interval. The normalized file sizes are achieved by dividing the compressed trajectories by the largest compressed trajectory among all methods for each task. Trajectories collected by MTC can be more efficiently compressed than baselines, which suggests that the trajectories produced by our policies show more repetitive, periodic structures to solve tasks. Furthermore, to more directly evaluate the predictability of policies, we trained a $t$-step-ahead action prediction model for different time steps $t$ in data sets that have been collected by policies learned with the different methods. Results in Appendix C.2 show that MTC achieves the smallest prediction errors among all methods for all time steps, indicating that actions produced by MTC are more easily predicted.

### 5.4. Non-periodic and High-dimensional Tasks

Our experiments on the DMC control tasks showed that total correlation regularization can improve the performance and robustness on locomotion tasks, which are characterized by periodic motions. As it is also interesting to investigate the performance of MTC also on task that require non-periodic behavior, we further evaluate our method on eight robotic manipulation tasks from the Metaworld benchmark (Yu et al., 2020). MTC achieves higher average rewards than baselines on all tasks (see Fig. 4 right and Fig. 8 in Ap-

pendix C.3). These results suggest that encouraging the agent to generate smooth and simple trajectories by maximizing total correlation contributes to successful completion of the manipulation task.

We further evaluate our approach on six image-based DMC tasks from the Planet benchmark (Hafner et al., 2019). On these tasks, the policy selects actions based on raw high-dimensional images rather than compact states. To handle image observations, we use a convolutional neural network encoder and a random shift augmentation. To ensure a fair comparison, each algorithm use the same encoder and image augmentation. We compare the aggregated performance across 6 tasks in Fig. 4 (left). MTC achieves better performance than leading baselines, including RPC, CURL (Laskin et al., 2020), SAC-AE (Yarats et al., 2021), and SAC. We refer to Appendix B.9 for more experimental details and Appendix C.4 for scores on individual tasks.

### 5.5. Ablation

Minimizing the KL divergence between the encoder and the dynamics model (the third term in Eq. 3) and the KL divergence between the policy and the action prediction model (the last term in Eq. 3) both help to induce predictable trajectories. To further investigate our regularizer, we evaluate the effect of these two KL divergences. Specifically, we consider two ablations of MTC, namely MTC-NoA which removes the KL divergence between the policy and the action prediction model in Eq. 3, and SAC which learns a policy without total correlation maximization. We evaluate MTC and its two ablations with respect to original task performance, robustness to state, action and dynamics changes on the Walker Stand task and the Cheetah Run task.

Fig. 5 shows the experimental results. Each subplot shows mean and 90% confidence interval from 30 episodes, averaged over 20 seeds. On the Walker Stand task, MTC achieves better performance than its two ablations in the presence of observation noise and strong dynamics perturbations. When action is perturbed by Gaussian noise, MTC achieves competitive performance compared to MTC-NoA but significantly outperforms SAC. On the Cheetah Run task, MTC achieves better robustness to action noise and mass changes, while being comparable to its two ablations in terms of robustness to observation noise. The improvements over MTC-NoA suggest that learning the action prediction model improves robustness. In addition, we also observe that overall MTC-NoA outperforms SAC, indicating that learning the dynamics model achieves robustness benefits.

We also evaluate the effect of the hyperparameter $I_p$ which is used for optimizing the weight $\alpha$ of the total correlation objective. Increasing the value of $I_p$ results in larger values of $\alpha$ and therefore biases the agent to increase total correlation. We evaluate the effect of $I_p$ with respect to

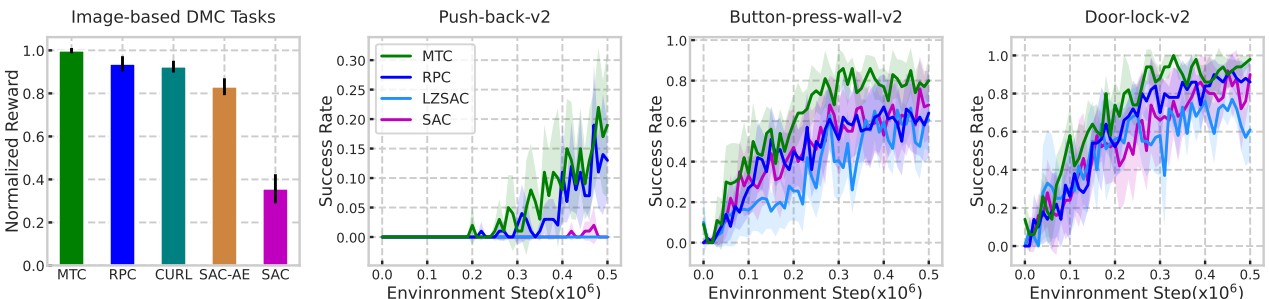

Figure 4. **Left:** aggregated performance of MTC and baselines at 500K environment steps on six image-based DMC tasks. The plot shows the normalized average rewards over 5 runs and 6 tasks, with error bars representing 90% confidence interval. For each run, we collect 10 evaluation episodes. MTC achieves better performance than baselines. **Right:** performance of our method and baselines on three manipulation tasks from Metaworld. The curves represent the average success rate over 10 different runs, with 90% confidence interval. Each run collects 10 evaluation episodes. MTC is competitive to baselines.

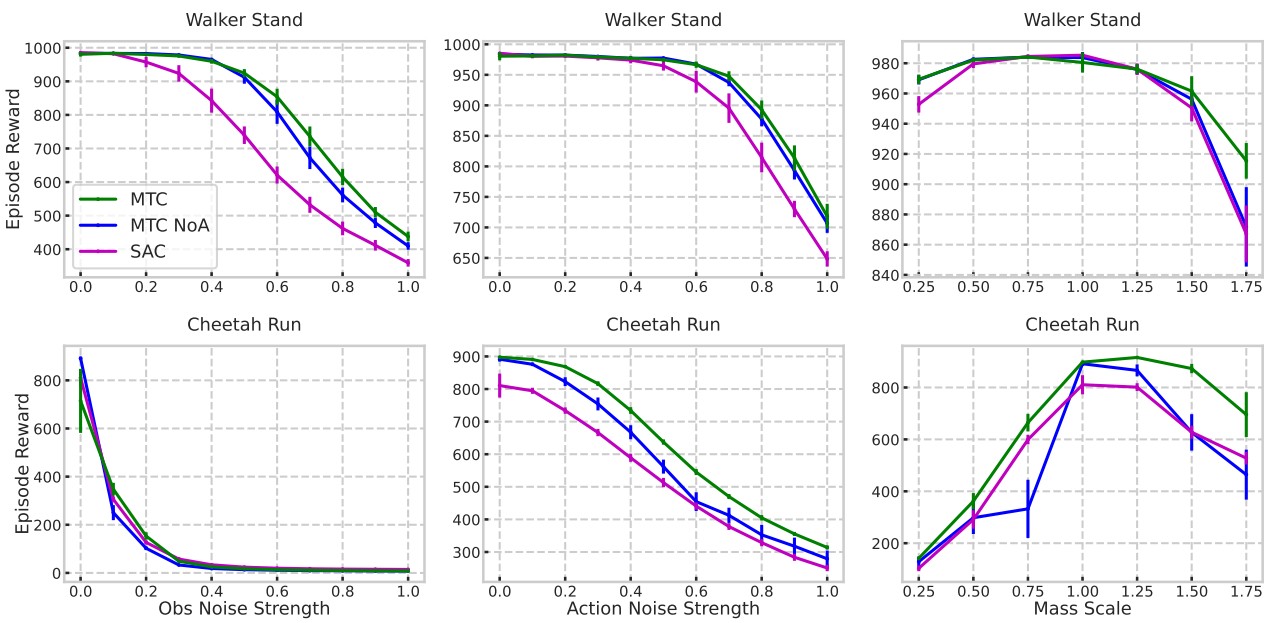

Figure 5. We test the robustness of MTC and its two ablations, MTC-NoA and SAC, on the Walker Stand task and the Cheetah Run task. Overall, MTC achieves better or at least comparable average rewards in the presence of observation noise (left column), action noise (middle column), and mass changes (right column) than its ablations.

original task performance, compressibility, and robustness to state, action, and dynamics perturbations on the Walker Stand task. Fig. 9 in Appendix C shows the experimental results. Each subplot shows mean and 90% confidence interval from 30 episodes, averaged over 20 seeds. We observe that tightening the lower bound of our total correlation objective by increasing $I_p$ doesn't hurt the final performance (rewards without perturbations) but significantly decreases the encoding size of trajectories. This suggests that maximizing the lower bound of the total correlation helps induce compressible or structured behaviors. We also find that in-

creasing $I_p$ effectively improves the robustness of learned policies to observation noise, action noise, and changes in dynamics (see Fig. 9). This supports our claim that biasing policies to focus on the essentials helps increase robustness to perturbations.

## 6. Discussions and Limitations

Our regularizer in Eq. 4 is related to the regularizer of RPC (Eysenbach et al., 2021), but generalizes it by considering the previous trajectory instead of only using the

information of the current step $t$, and by also including an action prediction model. These differences enable us to improve temporal consistency within trajectories, which significantly improves the consistency and robustness of the resulting behavior, as shown in our experiments. Furthermore, the regularizer in RPC was derived as the negative of an upper bound on the mutual information between raw and latent state sequences, $I(s_{1:T}; z_{1:T})$, whereas we prove in Appendix A.2, that our objective can not be derived from that perspective. We can, however, derive RPC from our formulation, showing that maximizing total correlation provides an important new perspective on regularization in reinforcement learning that not only results in more consistent and robust behavior, but also deepens our theoretical understanding of related works.

However, our lower bound of the total correlation corresponds to a sum of negated KL divergences, and is therefore always negative. Hence, it is not useful for estimating the actual total correlation, which we know to be positive. While a vacuous bound may not be useful for estimation, it can still be valuable for optimization, as in the case of subtracting a constant offset from the true objective. As demonstrated in our experiments, our lower bound is very effective for producing consistent behavior.

## 7. Conclusion and Future Work

Auxiliary objectives that create inductive biases towards simpler solutions (regularizers) are commonly, and very successfully, used in machine learning to learn more generalizable and robust solutions. We propose to use the total trajectory correlation as a novel regularizer for reinforcement learning, which acts on the level of the behavior. By directly corresponding to the information-theoretic compressibility of the induced trajectories, the total correlation is arguably the most principled choice to quantify the simplicity of a behavior. As directly maximizing the total correlation is intractable, we derived a variational lower bound and used it to formulate a regularized reinforcement learning problem that can be solved with standard techniques. Compared to similar sequence-based regularizers, total correlation regularization achieved very promising results by producing more consistent behavior that is more robust to state-, action- and dynamics perturbations. Hence, we believe that total trajectory correlation may serve as an important goal post for future reinforcement learning methods. Developing alternate bounds or approximations that better capture the total correlation while maintaining tractability is a promising direction for future research.

## Acknowledgement

Funded by the National Key Research and Development Program of China - Project number 2024YFB4708000, the German Research Foundation (DFG) - Project number PE 2315/18-1, and the German Federal Ministry of Research, Technology and Space (BMBFTR) - Project number 01IS23057B. This project has been supported by a hardware donation by NVIDIA through the Academic Grant Program.

## Impact Statement

This paper presents work whose goal is to advance the field of Machine Learning. There are many potential societal consequences of our work, none which we feel must be specifically highlighted here.

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

# A. Proofs

## A.1. Derivation Details of the Lower Bound

In this section, we provide full details about how to derive the lower bound (Eq. 2) from the total correlation definition. We start from the definition of the total correlation and derive a lower bound using a variational approximation $q(z_{1:T}, a_{1:T-1})$ of the trajectory distribution.

$$
\begin{aligned}
\mathcal{C}(z_1; a_1; \ldots; a_{T-1}; z_T) &= \mathbb{E}_{p(z_{1:T}, a_{1:T-1})} \left[ \log \frac{p(z_{1:T}, a_{1:T-1})}{\prod_{t=1}^{T} p(z_t) \prod_{t=1}^{T-1} p(a_t)} \right] \\
&= \mathbb{E}_{p(z_{1:T}, a_{1:T-1})} \left[ \log \frac{q(z_{1:T}, a_{1:T-1})}{\prod_{t=1}^{T} p(z_t) \prod_{t=1}^{T-1} p(a_t)} \right] \\
&\quad + \mathbb{D}_{\mathrm{KL}} \Big( p(z_{1:T}, a_{1:T-1}) \| q(z_{1:T}, a_{1:T-1}) \Big) \\
&\geq \mathbb{E}_{p(z_{1:T}, a_{1:T-1})} \left[ \log \frac{q(z_{1:T}, a_{1:T-1})}{\prod_{t=1}^{T} p(z_t) \prod_{t=1}^{T-1} p(a_t)} \right].
\end{aligned}
\tag{8}
$$

We parameterize the variational distribution $q(z_{1:T}, a_{1:T-1})$ autoregressively:

$$
q(z_{1:T}, a_{1:T-1}) = p(z_1) q(a_1|z_1) \prod_{t=1}^{T-1} q_\eta(z_{t+1}|z_{1:t}, a_{1:t}) q(a_{t+1}|z_{1:t+1}, a_{1:t}),
\tag{9}
$$

where $q_\eta(z_{t+1}|z_{1:t}, a_{1:t})$ is a history-based dynamics model, $q(a_{t+1}|z_{1:t+1}, a_{1:t})$ a history-based action model.

We plug Eq. 9 into Eq. 8, and then obtain

$$
\begin{aligned}
\mathcal{C}(z_1; a_1; \ldots; a_{T-1}; z_T) &\geq \mathbb{E}_{p(z_{1:T}, a_{1:T-1})} \left[ \log \frac{p(z_1) q(a_1|z_1) \prod_{t=1}^{T-1} q_\eta(z_{t+1}|z_{1:t}, a_{1:t}) q(a_{t+1}|z_{1:t+1}, a_{1:t})}{\prod_{t=1}^{T} p(z_t) \prod_{t=1}^{T-1} p(a_t)} \right] \\
&= \mathbb{E}_{p(z_{1:T}, a_{1:T-1})} \left[ \log \frac{\prod_{t=1}^{T-1} q_\eta(z_{t+1}|z_{1:t}, a_{1:t})}{\prod_{t=1}^{T-1} p(z_{t+1})} \right] \\
&\quad + \mathbb{E}_{p(z_{1:T}, a_{1:T-1})} \left[ \log \frac{\prod_{t=1}^{T-1} q_\chi(a_t|z_{1:t}, a_{1:t-1})}{\prod_{t=1}^{T-1} p(a_t)} \right] \\
&= \mathbb{E}_{p(z_{1:T}, a_{1:T-1})} \left[ \sum_{t=1}^{T-1} \log \frac{q_\eta(z_{t+1}|z_{1:t}, a_{1:t})}{p(z_{t+1})} \right] \\
&\quad + \mathbb{E}_{p(z_{1:T}, a_{1:T-1})} \left[ \sum_{t=1}^{T-1} \log \frac{q_\chi(a_t|z_{1:t}, a_{1:t-1})}{p(a_t)} \right].
\end{aligned}
\tag{10}
$$

The marginal distributions $p(z_{t+1})$ and $p(a_t)$ are unknown. However, the conditional distributions $f_\theta(z_{t+1}|s_{t+1})$ and $\pi_\phi(a_t|s_t)$ are known and can be substituted while maintaining a lower bound:

$$
\begin{aligned}
&\mathcal{C}(z_1; a_1; \ldots; a_{T-1}; z_T) \geq \\
&\mathbb{E}_{p(z_{1:T}, a_{1:T-1})} \left[ \sum_{t=1}^{T-1} \log \frac{q_\eta(z_{t+1}|z_{1:t}, a_{1:t})}{f_\theta(z_{t+1}|s_{t+1})} \right] + \mathbb{E}_{p(s_{1:T}, z_{1:T}, a_{1:T-1})} \left[ \sum_{t=1}^{T-1} \log \frac{q_\chi(a_t|z_{1:t}, a_{1:t-1})}{\pi_\phi(a_t|s_t)} \right] \\
&+ \sum_{t=1}^{T-1} \mathbb{E}_{p(s_{t+1})} \left[ \mathbb{D}_{\mathrm{KL}} \Big( f_\theta(z_{t+1}|s_{t+1}) \| p(z_{t+1}) \Big) \right] + \sum_{t=1}^{T-1} \mathbb{E}_{p(s_t)} \left[ \mathbb{D}_{\mathrm{KL}} \Big( \pi_\phi(a_t|s_t) \| p(a_t) \Big) \right] \\
&\geq \mathbb{E}_{p(z_{1:T}, a_{1:T-1})} \left[ \sum_{t=1}^{T-1} \log \frac{q_\eta(z_{t+1}|z_{1:t}, a_{1:t})}{f_\theta(z_{t+1}|s_{t+1})} \right] + \mathbb{E}_{p(s_{1:T}, z_{1:T}, a_{1:T-1})} \left[ \sum_{t=1}^{T-1} \log \frac{q_\chi(a_t|z_{1:t}, a_{1:t-1})}{\pi_\phi(a_t|s_t)} \right]
\end{aligned}
\tag{11}
$$

where the inequality in the last line holds because of the non-negativity of the KL divergence. Here we obtain the lower bound in Eq. 2. Notably, the first term and the second term in the last line of Eq. 11 both are lower bounds on the total correlation.

**A.2. Connections to $I(s_{1:T}; z_{1:T})$**

RPC (Eysenbach et al., 2021) aims to minimize the following upper bound of the mutual information between the state sequence and the latent state sequence,

$$I(s_{1:T}; z_{1:T}) = \mathbb{E}_{p(s_{1:T}, z_{1:T})}\left[\log \frac{p(z_{1:T}|s_{1:T})}{p(z_{1:T})}\right] \leq \mathbb{E}_{p(s_{1:T}, z_{1:T}, a_{1:T})}\left[\log \frac{\prod_{t=1}^{T-1} f(z_{t+1}|s_{t+1})}{\prod_{t=1}^{T-1} q(z_{t+1}|z_t, a_t)}\right]. \tag{12}$$

In contrast to our bound, this bound does not use the history for the dynamics model, and it does not explicitly account for action consistency. Furthermore, we argue that the lower bound (Eq. 12) does not always hold as it was derived by replacing $p(z_{1:T}|s_{1:T})$ by $\prod_{t=1}^{T-1} p(z_{t+1}|s_{t+1})$ (Eysenbach et al., 2021, Appendix C1). These distributions are in general not the same because information about future state observations can decrease uncertainty about the current latent state, and therefore

$$p(z_{t+1}|z_{1:t}, s_{1:T}) \neq p(z_{t+1}|s_{t+1}).$$

We will now show that the latter replacement may invalidate the upper-bound by analyzing the gap,

$$\mathbb{E}\left[\log \frac{p(z_{1:T}|s_{1:T})}{p(z_{1:T})}\right] - \mathbb{E}\left[\log \frac{\prod_{t=1}^{T-1} f(z_{t+1}|s_{t+1})}{\prod_{t=1}^{T-1} q(z_{t+1}|z_t, a_t)}\right]$$

$$= \mathbb{E}_{p(s_{1:T})}\underbrace{\left[\mathbb{D}_{\text{KL}}\Big(p(z_{1:T}|s_{1:T})||p(z_{1:T})\Big)\right.}_{\geq 0}$$

$$\underbrace{\left. - \sum_{t=1}^{T-1} \mathbb{E}_{p(s_{t+1}, a_{1:t}, z_{1:t})}\left[\mathbb{D}_{\text{KL}}\Big(f(z_{t+1}|s_{t+1})||q(z_{t+1}|z_t, a_t)\Big)\right]\right].}_{\geq 0}$$

The second term, may in general be smaller than the first term, for example, when the variational distribution perfectly matches the encoder, and, thus, the second term does not upper-bound the mutual information $I(s_{1:T}, z_{1:T})$.

We can, however, derive RPC based on our total correlation perspective by using the variational distribution

$$q'(z_{1:T}, a_{1:T-1}) = p(z_1)p(a_1) \prod_{t=1}^{T-1} q_\eta(z_{t+1}|z_t, a_t)p(a_{t+1}) \tag{13}$$

instead of Eq 9.

**A.3. Updating Q function**

Following the standard recursive Bellman equation, the Q function with parameters $\upsilon$ can be optimized by minimizing the loss

$$L(\upsilon) = \mathbb{E}_{\mathcal{D}, f, \pi}\left[\left(Q_\upsilon(s_t, a_t) - y(s_t, a_t)\right)^2\right] \tag{14}$$

where the target is given by

$$y(s_t, a_t) = r^*(s_t, a_t, s_{t+1}) + \gamma(1-d)\big[Q_\upsilon(s_{t+1}, a_{t+1}) - \beta \log(\pi_\phi(a_t|s_{t+1})\big] \tag{15}$$

with discounted factor $\gamma$ and termination flag $d$ and next action $a_{t+1}$ sampled from the current policy. We employ the independent target Q function to computer the target and stop the gradient through the target Q function.

**A.4. Total Correlation for Infinite Sequences**

The standard definition of total correlation is only useful for finite sequences, as it would typically not converge in the limit of infinite sequences. We will now show that our practical implementation which is based on an infinite-horizon MDP, maximizes a natural extension of the total correlation to infinite sequences.

We start by introducing the *per-step contribution of step $t$ to the total correlation*,

$$\mathcal{C}_t(x_1; \ldots; x_t) := \mathbb{E}_{x_1, x_2, \ldots, x_t} \left[ \log \frac{p(x_t | x_1, \ldots, x_{t-1})}{p(x_t)} \right],$$

and note that for any finite sequence, its total correlation corresponds to the sum of per-step-contributions,

$$\mathcal{C}(x_1; x_2; \ldots; x_n) = \mathbb{E}_{x_1, x_2, \ldots, x_n} \left[ \log \frac{p(x_1, x_2, \ldots, x_n)}{\prod_{i=1}^n p(x_i)} \right].$$
$$= \sum_{t=1}^N \mathcal{C}_t(x_1; \ldots; x_t).$$

Intuitively, $\mathcal{C}_t(x_1; \ldots; x_t)$ corresponds to the additional amount of information (measured in nats) that we can save (compared to per-step encodings) when encoding a sequence of length $t$ instead of a sequence of length $t - 1$,

$$\mathcal{C}_t(x_1; \ldots; x_t) = \mathcal{C}(x_1; \ldots; x_t) - \mathcal{C}(x_1; \ldots; x_{t-1}).$$

.

Using the per-step contribution of the total correlation $\mathcal{C}_t$, we define the discounted total correlation $\mathcal{C}^\gamma(x_1; \ldots; x_\infty)$ of an infinite sequence $(x_1, \ldots, x_\infty)$ as

$$\mathcal{C}^\gamma(x_1; \ldots; x_\infty) := \sum_{t=0}^\infty \gamma^t \mathcal{C}_t(x_1; \ldots; x_t). \tag{16}$$

Hence, the discounted total correlation corresponds to the total correlation, when geometrically discounting future per-step contributions.

Using this definition, it can be shown analogously to the derivations in Appendix A.1, that maximizing our discounted objective 5 maximizes a lower bound of the discounted total correlation $\mathcal{C}^\gamma(x_1; \ldots; x_\infty)$.

# B. Experimental Details

## B.1. Task Specification

We test our algorithms on MuJoCo-powered continuous control tasks from the Deepmind Control, which provides a standardized set of benchmarks for reinforcement learning agents. For each task, the episode length is set to 1000 steps, and the action vector is bounded into $[-1, 1]$. We refer to (Tassa et al., 2018) for more descriptions of tasks.

## B.2. Implementation Details

**SAC codebase.** We implement our algorithm on top of the common PyTorch implementation of the SAC algorithm (Yarats et al., 2021). We used the default hyperparameters from that implementation unless specified otherwise. Detailed descriptions of the SAC implementation are available in (Yarats et al., 2021).

**Encoder.** The encoder $f_\theta(z_t|s_t)$ is parametrized as a 3-layer neural network with FCN (units=256) $\rightarrow$ FCN (units=256) $\rightarrow$ FCN (units=60) architecture and ReLU hidden activations. Its output is divided into the mean and the standard deviation of a diagonal Gaussian distribution.

**Prediction models.** Our prediction models $q_\eta(z_{t+1}|z_{1:t}, a_{1:t})$ and $q_\eta(a_t|z_{1:t}, a_{1:t-1})$ are parameterized by an LSTM module followed by a 3-layer neural network. The LSTM module is implemented using the common nn.LSTM class provided by PyTorch. The hidden dimension is set to 256, the output dimension is set to 30, and the number of recurrent layers is set to 1 for the LSTM module. The 3-layer neural network has the same architecture and activation function as the encoder. The output of the dynamic model is normalized and then divided into the mean and the standard deviation of a diagonal Gaussian distribution.

**Dual multipliers.** We treat the hyperparameter $\alpha$ as a dual multiplier and optimize it via dual gradient ascent. We initialize the value of $\alpha$ to $10^{-6}$ and parametrize it as $\log \alpha$ to ensure that it remains positive during optimization. For optimizing the entropy coefficient, we take the contribution from $\alpha$ into account and directly optimize $\beta' = \beta + \alpha$.

**Bound optimization.** We refer to the first term and the second term in the last line of Eq. 11 as the state bound $\widetilde{\mathcal{C}}_z$ and the action bound $\widetilde{\mathcal{C}}_a$, respectively. In practice, we implement our lower bound $\widetilde{\mathcal{C}}$ in Eq. 2 as a weighted combination of the state and action bounds, $(1 - m)\widetilde{\mathcal{C}}_z + m\widetilde{\mathcal{C}}_a$ with an additional coefficient $m \in [0, 1]$. Since the state bound and the action bound both are lower bounds on total correlation (see Eq. 11), their weighted combination is still a lower bound on total correlation.

## B.3. Other Hyperparameters

We initialize the replay buffer with 5000 samples from the initial policy and train all agents for 1 million steps. We evaluate the agent every 20000 steps. All learnable parameters are updated using the Adam optimizer with a learning rate of $10^{-4}$. We determine information constraints $I_p$ and history length by performing hyperparameter tuning. We provide an overview of our used hyperparameters in Table. 2. For other details, please refer to the provided code.

## B.4. Extended Description of Baseline Implementations

**SAC.** We obtain the results for SAC by running the PyTorch implementations of SAC (Yarats et al., 2021). We use the same hyperparameters for SAC as our algorithm to ensure a fair comparison. We found that our obtained results for SAC are stronger than the results of SAC reported in previous work (Yarats & Kostrikov, 2020).

**LZ-SAC.** We use the official implementation provided by Saanum et al. (2023) to obtain the results for LZ-SAC, since the official implementation is based on the same codebase of SAC and the hyperparameters has been tuned to achieve good results on DMC tasks.

**RPC.** To obtain the results for RPC, we first use the original code provided by Eysenbach et al. (2021), which is built on top of the SAC implementation from TF-Agents. To achieve as good performance as possible for RPC, we perform hyperparameter tuning to select the suitable information constraint for RPC. To ensure a fair comparison, we additionally implement RPC by ourselves, using the same codebase of SAC as MTC and LZ-SAC. We use the same SAC hyperparameters for our implementation of RPC as our algorithm.

*Table 2.* Hyperparameters used in MTC.

| Parameter | Value |
|---|---|
| $I_p$ for Cheetah Run, Hopper and Walker Stand | -0.5 |
| $I_p$ for other tasks | -7.0 |
| history length | 8 |
| Replay buffer capacity | 1 000 000 |
| Optimizer | Adam |
| Critic learning rate | $10^{-4}$ |
| Critic Q-function soft-update rate | 0.01 |
| Critic target update frequency | 2 |
| Actor learning rate | $10^{-4}$ |
| Actor update frequency | 1 |
| Actor log stddev bounds | [-10 2] |
| Temperature learning rate | $10^{-4}$ |
| Initial temperature | 0.1 |
| Initial steps | 5000 |
| Discount | 0.99 |
| Initial $\alpha$ | $10^{-6}$ |
| $\alpha$ learning rate | $10^{-4}$ |
| Representation dimension | 30 |
| Number of training steps | $10^6$ |
| Coefficient $m$ | $10^{-6}$ |
| Batch Size for Hopper and Acrobot | 512 |
| Batch Size for other tasks | 256 |

### B.5. Robustness to Observation Noise

In all experiments of robustness and trajectory compression, for each agent, we evaluated the performance of policies saved after finishing the training for 1M steps. Gaussian noise is regarded as a strong state distractor for reinforcement learning algorithms in prior work (Bai et al., 2021). We add the Gaussian noise to observations and the learned policies select the actions based on the noisy observations.

### B.6. Robustness to Action Noise

Noise added to actions can be viewed as a type of environment perturbation. In this experiment, we first use the saved policies to select the action based on the current state, and add the Gaussian noise to the chosen action. We then clip the action into $[-1, 1]$ before passing the action signal to the task.

### B.7. Robustness to Dynamics

We modify the mass of the robot body to test the robustness of learned policies. In our experiment setup, we get the body mass of the robot via the env.physics.model.body_mass attribute provided by the environment. Since the body mass varies across different tasks, we change the mass by scaling it, rather than increasing or decreasing a constant. We then evaluate the performance of the learned policies on the environment with the changed mass.

### B.8. Trajectory Consistency

In our experiment, we measure the compressibility of trajectories using the bzip2 algorithm, which is easily available by installing the common bz2 python package. For each seed, we collect 30 trajectories using learned policies. Since the collected trajectories have the same number of data points and these data points have the same numerical precision, the uncompressed trajectories collected by the different algorithms have the same file size. We compress individual trajectories by calling the bz2.compress() method provided by the bz2 package. Smaller file sizes of compressed trajectories mean that trajectories can be better compressed.

## B.9. Image-based DMC Tasks

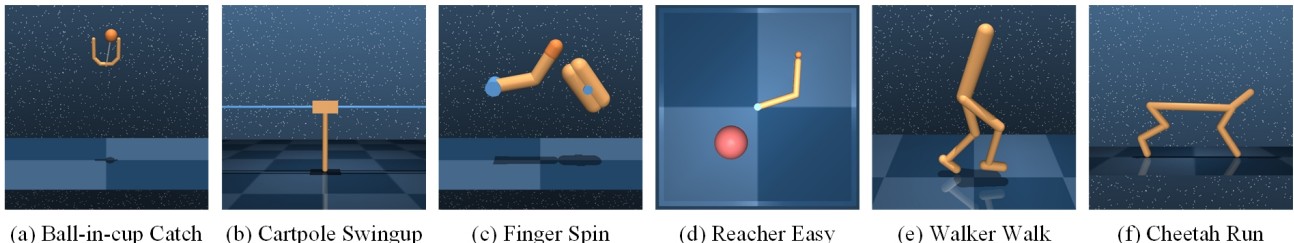

(a) Ball-in-cup Catch    (b) Cartpole Swingup    (c) Finger Spin    (d) Reacher Easy    (e) Walker Walk    (f) Cheetah Run

*Figure 6.* We evaluate our method on eight image-based DMC tasks.

We evaluate the performance of our method on the commonly used PlaNet benchmark, which consists of a set of complex image-based continuous control tasks. Specifially, we consider six tasks:Ball-in-cup Catch, Cartpole Swingup, Finger Spin, Reacher Easy, Walker Walk, and Cheetah Run (see Figure 6). The flexibility of MTC allows us to apply it to image-based control settings with minimal modifications. We employ the convolutional encoder architecture from SAC-AE (Yarats et al., 2021) for encoding raw images into representations. Following common practice, we perform data augmentation by randomly shifting the image by $[-4, 4]$, before we feed images into the encoder. We obtain an individual observation by stacking 3 consecutive frames, where each frame is an RGB rendering image with size $84 \times 84 \times 3$ from the 0th camera. An overview of our used hyperparameters for imaged-based tasks is shown in Table. 3. We refer to our code for more implementation details.

*Table 3.* Hyperparameters used in Image-based tasks.

| Parameter | Value |
|---|---|
| $I_p$ for Finger and Ball-In-Cup | -10.0 |
| $I_p$ for other tasks | -0.1 |
| history length for Finger and Ball-In-Cup | 3 |
| history length for other tasks | 5 |
| Replay buffer capacity | 1 00 000 |
| Optimizer | Adam |
| Critic learning rate | $10^{-4}$ |
| Critic Q-function soft-update rate | 0.01 |
| Critic target update frequency | 2 |
| Actor learning rate | $10^{-4}$ |
| Actor update frequency | 1 |
| Actor log stddev bounds | [-10 2] |
| Temperature learning rate | $10^{-4}$ |
| Initial temperature | 0.1 |
| Initial steps | 5000 |
| Discount | 0.99 |
| Initial $\alpha$ | $10^{-6}$ |
| $\alpha$ learning rate | $10^{-4}$ |
| Representation dimension | 50 |
| Coefficient $m$ | 0.0001 |
| Batch Size | 256 |

## B.10. Compute Resources

We performed every experiment on an Intel(R) Xeon(R) E5-2620 CPU with GeForce GTX 2080 Ti graphics card and used approximately one day for training.

# C. Additional Results

## C.1. Learning Curves

We show the learning curves of MTC and other approaches on eight DMC tasks in Fig. 7. MTC achieves higher rewards and faster learning speed than baselines on the majority of tasks. Besides, MTC has a good learning stability on all tasks.

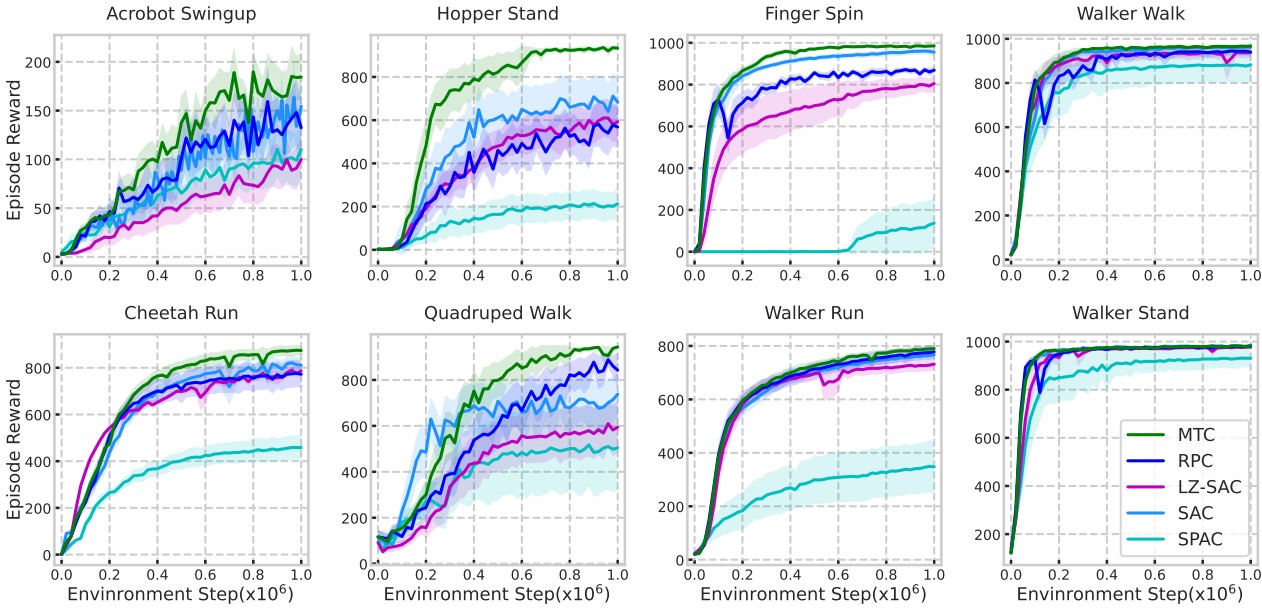

*Figure 7.* Learning curves of our method and baselines on 8 DMC tasks. The plot shows the average rewards over 20 seeds with a shading of 90% confidence interval. MTC achieves better or at least comparable performance and sample efficiency than baselines.

## C.2. Predictability of Policies

To directly assess the correlation of learned policies, we additionally trained a $t$-step-ahead predictive model for $t = [3, 5, 8, 10]$ time steps on data sets that have been collected with the policy that has been learned with the different methods for a locomotion task and a manipulation task. The model was trained to predict the action $t$ steps ahead, when given the current action. We train the predictive model using maximum likelihood estimation. Table 4 and Table 5 compare the prediction errors of the learned predictive models on the Finger Spin and Drawer-open-v2, respectively. For all tested time differences, the prediction error of MTC was the smallest among all baselines.

*Table 4.* We compare the prediction errors (negative log-likelihood) of the learned predictive model for 4 different $t$ time steps on the Finger Spin task. MTC achieves smaller prediction errors than other methods on all time steps.

| Prediction Error (negative log-likehood) | t=3 | t=5 | t=8 | t=10 |
| --- | --- | --- | --- | --- |
| MTC | **0.22** | **0.32** | **0.48** | **0.67** |
| RPC | 0.34 | 0.58 | 0.72 | 0.76 |
| LZ-SAC | 0.96 | 1.31 | 1.48 | 1.53 |
| SAC | 0.78 | 1.13 | 1.47 | 1.57 |

## C.3. Metaworld Tasks

We compare MTC to strong baselines, RPC and SAC, on 5 manipulation tasks from Metaworld in Fig 8. Overall, MTC achieves higher mean rewards than RPC and SAC on all manipulation tasks.

*Table 5.* We compare the prediction errors (negative log-likelihood) of the learned predictive model for 4 different $t$ time steps on the Drawer-open-v2 from Metaworld. MTC achieves smaller prediction errors than other methods on all time steps.

| Prediction Error (negative log-likehood) | t=3 | t=5 | t=8 | t=10 |
|---|---|---|---|---|
| MTC | **-4.84** | **-4.40** | **-4.05** | **-3.35** |
| RPC | -2.62 | -3.05 | 0.14 | 0.17 |
| SAC | -2.37 | -2.07 | -2.7 | -1.7 |

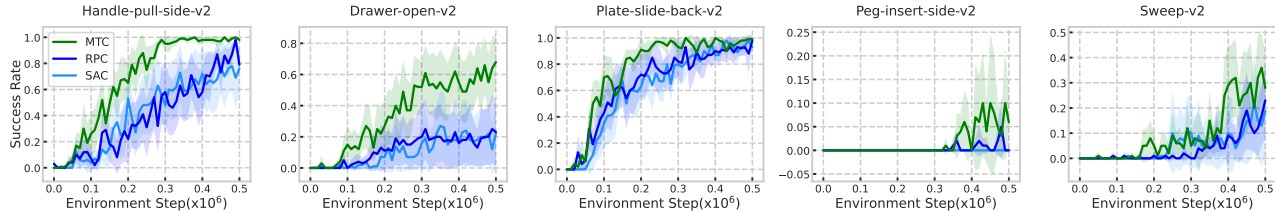

*Figure 8.* Performance of our method and baselines on 5 manipulation tasks from Metaworld. The plot shows the average success rate and 90% confidence interval over 10 seeds. MTC achieves comparable performance than baselines.

## C.4. Image-based DMC Tasks

Table. 6 shows the performance of our approach and baselines on six image-based DMC tasks. Overall, MTC achieves higher average rewards than baselines on five of six tasks. On the Walker Walk task, for instance, MTC achieves a reward of 939, higher than 917 achieved by RPC and 897 achieved by CURL.

*Table 6.* Scores (means and standard error over 5 seeds) achieved by our method and baselines on six DMC tasks at 500K environment steps. Each run includes 10 evaluation episodes. MTC achieves higher or at least comparable average reward than all baselines.

| Scores | MTC | RPC | CURL | SAC-AE | SAC |
|---|---|---|---|---|---|
| Cartpole Swingup | **868 ± 1** | **859 ± 8** | 837±15 | 748±47 | 436±94 |
| Ball-in-cup Catch | 956± 3 | **964 ± 3** | **957± 6** | 831± 25 | 355±77 |
| Finger Spin | **979± 2** | 695 ± 43 | 854 ± 48 | 839± 68 | 530±24 |
| Walker Walk | **939± 6** | 917± 15 | 897 ± 26 | 836± 24 | 97±62 |
| Reacher Easy | **968± 6** | 938± 22 | 891 ± 30 | 678± 61 | 191±40 |
| Cheetah Run | **585 ± 25** | **572 ± 15** | 492± 22 | 476± 22 | 250±26 |

## C.5. Ablations on History-based Policies

While our total correlation objective results in a reward function that depends on the past in order to encourage the agent to perform actions that are consistent with its previous actions, we did not choose a history-based policy in our experiments for a simpler and fairer comparison. We ran additional experiments to test the effect of history-based observations for standard SAC and our method on the Finger Spin task. The results are consistent with non-history-based policies (see Table. 7), indicating that the improvements are caused by our total correlation objective, whereas non-markovianity on observations is not sufficient.

*Table 7.* Performance comparison between MTC and SAC and their ablations, whose policies use history-based observations. The table shows the mean and 90% confidence interval over 10 seeds. For each run, we collected episode rewards at 500K environment steps and computed the mean over 10 evaluation episodes. MTC achieves consistent performance with its history-based ablation, MTC-history.

| Finger Spin | MTC | MTC-history | SAC | SAC-history |
|---|---|---|---|---|
| Score | **985 ± 2** | **986 ± 3** | 955 ± 18 | 951 ±23 |

## C.6. Ablations on Hyperparameters

We evaluate the performance of MTC for three different constraints $I_p$. The robustness to mass changes and observation and action noise, and the compression of behaviors are improved while increasing $I_p$.

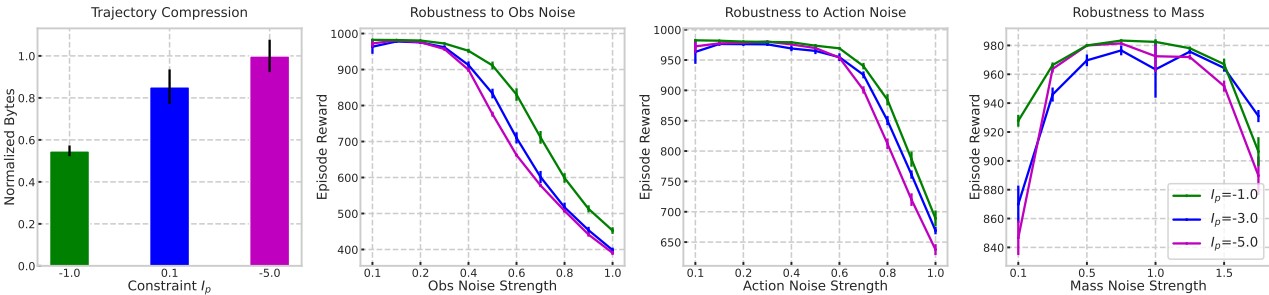

*Figure 9.* We test the performance of MTC with different constraints $I_p$. The robustness to mass changes and observation noise, and the compression of behaviors are improved while increasing $I_p$.

## C.7. Robustness Comparison on A Single Task

A constant increase in noise could result in moderate effects relative to the zero-noise level, larger effects when applied on top of some existing noise (due to compounding effects), and in negligible effects when applied to very large noise level where the policy is not able to perform reasonably anyway. Hence, it is difficult to compare the robustness to noise when the different policies already start at different reward levels. However, we noticed that MTC, RPC, and SAC perform very similarly on Walker-Stand and investigated the robustness while focusing on this single environment. As shown in Fig. 10, the robustness of MTC increases with respect to observation noise, action noise as well as mass change.

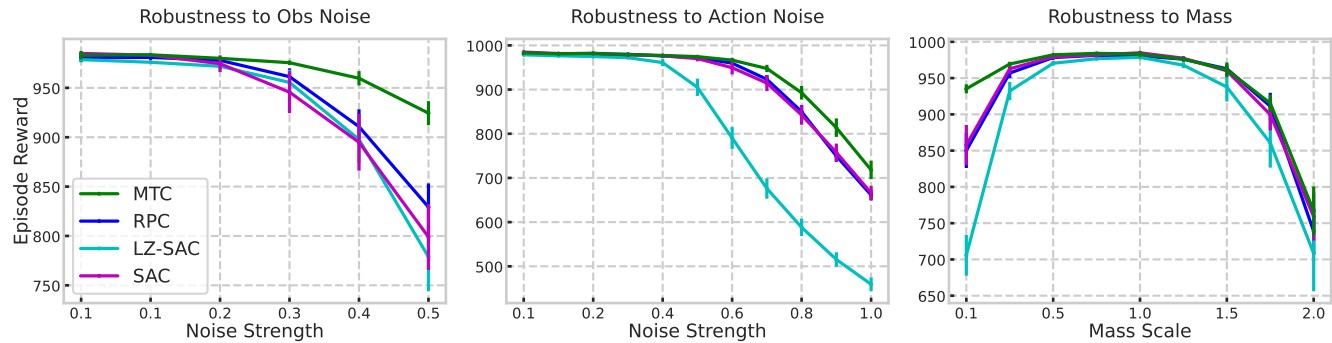

*Figure 10.* Robustness to observation noise (left), action noise (middle), mass changes (right) on Walker Stand. The plot shows the average reward over 20 seeds, with error bars representing 90% confidence interval. MTC achieves higher rewards than baselines in the presence of strong observation, action, and mass perturbations.

## C.8. Robustness on Metaworld Tasks

We also evaluate the robustness to observation noise, action noise, and mass changes in eight manipulation tasks from Metaworld. Overall, MTC achieves better performance when actions and dynamics are perturbed, while being comparable to baselines in the presence of observation noise.

## C.9. Robustness to Spurious Correlation

We perform experiments to investigate if MTC is robust to spurious correlations. In the experiment, we added additional state dimensions that are not controllable by the actor, but instead follow a fixed Gaussian transition model. These distractors

*Table 8.* Scores (mean and 90% confidence interval for 20 runs) achieved by our method and baselines at eight manipulation tasks from Metaworld with action noise, mass changes, and observation noise. Overall, MTC obtained higher scores in the presence of action and dynamics perturbations, while being comparable to baselines on tasks with observation noise.

| 500K step scores | | MTC(Ours) | RPC | SAC |
|---|---|---|---|---|
| Action noise with strength 2.0 | Handle-pull-side-v2 | **0.98± 0.01** | 0.72 ± 0.05 | 0.62± 0.12 |
| | Drawer-open-v2 | **0.60± 0.13** | 0.21 ± 0.13 | 0.19± 0.10 |
| | Plate-slide-back-v2 | **0.71± 0.06** | **0.67 ± 0.08** | 0.58± 0.07 |
| | Peg-insert-side-v2 | **0.01± 0.01** | 0.00 ±0.00 | 0.00±0.00 |
| | Sweep-v2 | **0.01± 0.01** | 0.00 ±0.00 | 0.00±0.00 |
| | Button-press-wall-v2 | 0.24± 0.04 | **0.26 ± 0.05** | **0.31± 0.06** |
| | Door-lock-v2 | **0.84± 0.03** | 0.69 ± 0.06 | 0.71±0.06 |
| | Push-back-v2 | **0.10± 0.04** | **0.08 ± 0.03** | 0.01 ± 0.01 |
| Mass changes with scale 1.75 | Handle-pull-side-v2 | **0.98± 0.02** | 0.71 ± 0.07 | 0.65± 0.12 |
| | Drawer-open-v2 | **0.62± 0.13** | 0.22 ± 0.16 | 0.21± 0.15 |
| | Plate-slide-back-v2 | **0.98± 0.01** | 0.87 ± 0.08 | 0.93± 0.03 |
| | Peg-insert-side-v2 | **0.07± 0.06** | 0.00 ±0.00 | 0.00±0.00 |
| | Sweep-v2 | **0.29± 0.10** | 0.18 ± 0.09 | 0.18± 0.12 |
| | Button-press-wall-v2 | 0.51± 0.08 | 0.59 ± 0.12 | **0.62± 0.10** |
| | Door-lock-v2 | **0.96± 0.02** | 0.84 ± 0.05 | 0.86±0.05 |
| | Push-back-v2 | **0.18± 0.07** | **0.12 ± 0.04** | 0.00 ± 0.00 |
| Obs noise with strength 0.05 | Handle-pull-side-v2 | **0.95± 0.04** | 0.74 ± 0.07 | 0.69± 0.13 |
| | Drawer-open-v2 | **0.45± 0.15** | 0.18 ± 0.11 | 0.13± 0.08 |
| | Plate-slide-back-v2 | 0.29± 0.09 | 0.29 ± 0.08 | **0.46± 0.07** |
| | Peg-insert-side-v2 | **0.00± 0.00** | **0.00 ± 0.00** | **0.00± 0.00** |
| | Sweep-v2 | **0.02± 0.01** | **0.02 ± 0.02** | 0.01± 0.01 |
| | Button-press-wall-v2 | **0.27± 0.04** | 0.24 ± 0.08 | 0.16± 0.07 |
| | Door-lock-v2 | **0.76± 0.07** | 0.74 ± 0.04 | 0.73± 0.04 |
| | Push-back-v2 | **0.00± 0.00** | **0.00 ± 0.00** | **0.00± 0.00** |

are not correlated with the remaining states, the actions nor the reward that the agent receives, although by coincidence, it might appear that such correlations exists, resulting in spurious correlations. We evaluate our method and baselines on the Walker Stand task with such observation distractors. Fig. 11 shows the performance of our method, RPC and SAC. The plot shows the mean over 10 seeds, with a 90% confidence interval. MTC significantly outperforms RPC and SAC in rewards, suggesting that MTC improves robustness to spurious correlations.

### C.10. Visualizations of Trajectories

We visualize the trajectories produced by our method and baselines on the Finger Spin task. For the Finger Spin task, the dimensions of the action and state space are 2 and 9, respectively. Fig 12 and Fig 13 visualize the action and state trajectories produced by our method and baselines. We observed that MTC produces more consistent and periodic patterns in trajectories.

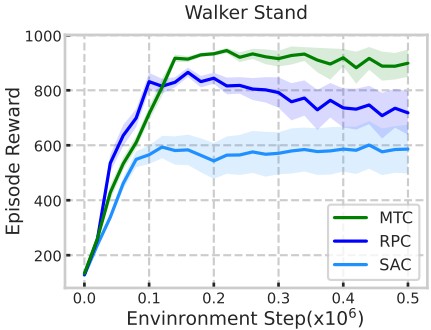

*Figure 11.* Robustness to spurious correlations on Walker Stand. MTC achieves higher rewards than RPC and SAC, when states are expanded with unrelated Gaussian noises.

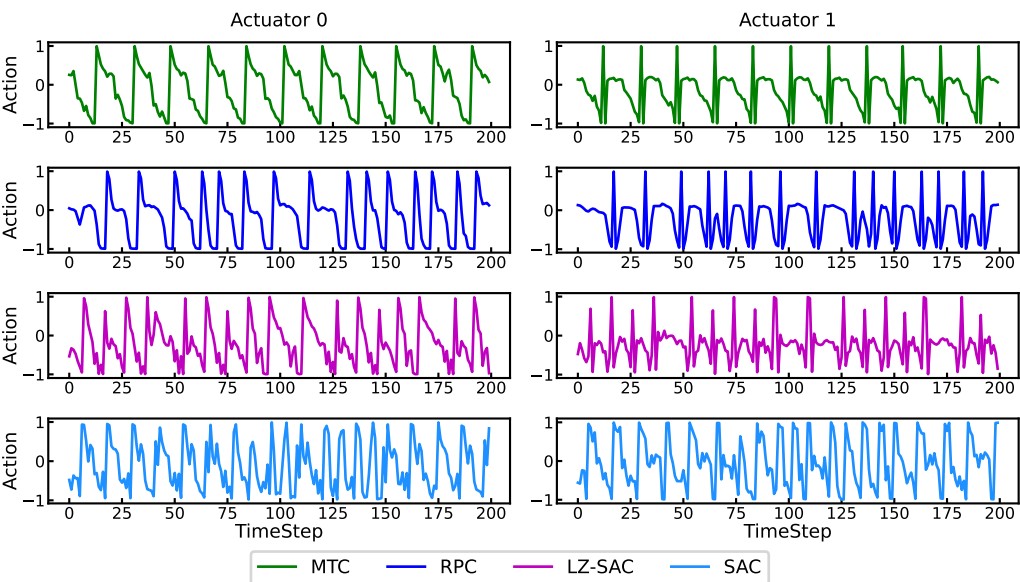

*Figure 12.* Visualizations of action sequences generated by our method and baselines on the Finger Spin task. MTC produces more consistent and periodic behavior than baselines.

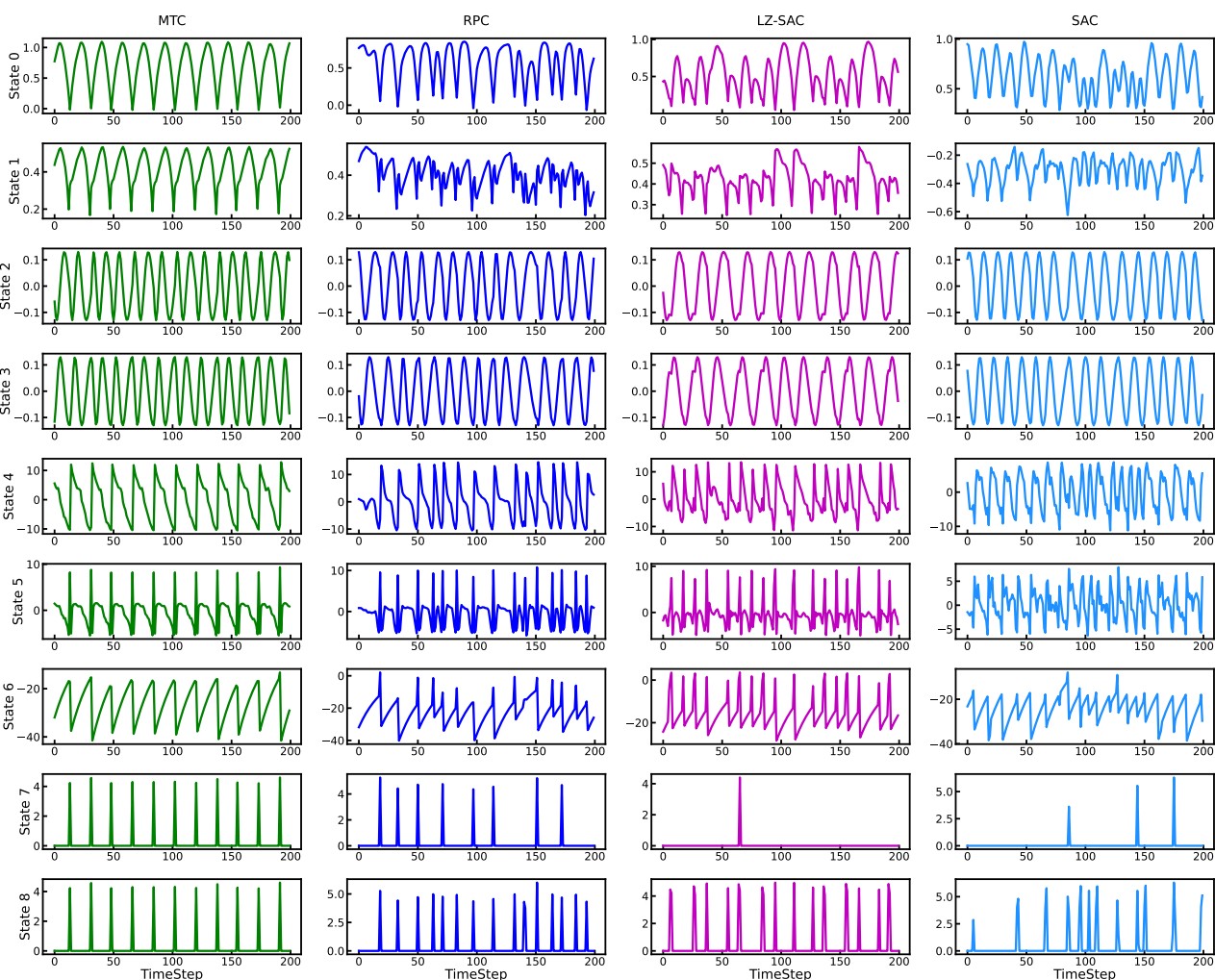

*Figure 13.* Visualizations of state sequences generated by our method and baselines on the Finger Spin task. State sequences of our method show more repeating and periodic patterns.

