# OpenReview forum: "Maximum Total Correlation Reinforcement Learning"
_ICML.cc/2025/Conference — ICML 2025 poster_

### Official Review · Reviewer_vAa9 · 2025-03-10

**Overall Recommendation:** 3

**Summary:**

This paper proposes an algorithm called MTC, which is a SAC-style approach that utilizes total correlation regularization. The basic idea behind MTC is that reducing unnecessary variations in states and actions increases robustness; accordingly, the authors propose an algorithm that maximizes the total correlation—a generalized mutual information among all states and actions within an episode. For the total correlation regularization term, a variational lower bound is derived, and MTC employs this lower bound as the regularization term. Experimental results demonstrate that MTC exhibits higher performance compared to other SAC-based algorithms and is particularly beneficial for enhancing robustness.

**Claims And Evidence:**

The main claim of this paper is that reducing unnecessary variations in states and actions enhances robustness. This claim is supported by various experiments. In particular, the paper presents graphs showing that maximizing the proposed total correlation increases the consistency of the state and action trajectories of the learned policy, and it provides results demonstrating robust performance against action noise, state noise, and mass variations, thereby offering persuasive evidence for the main claim.

**Essential References Not Discussed:**

An approach with a similar motivation is the use of Fourier transform, which has been explored in previous studies [1, 2]. These methods also aim to remove unnecessary information from state and action sequences to facilitate faster learning, which is aligned with the motivation of MTC. Explaining the differences between these studies and MTC, and incorporating additional experiments, could further enhance the novelty of the paper.

[1] A. Li & D. Pathak, “Functional regularization for reinforcement learning via learned fourier features,” NeurIPS 2021.
[2] M. Ye, et al., “State sequences prediction via fourier transform for representation learning,” NeurIPS 2023.

**Experimental Designs Or Analyses:**

1. In Figure 2, the magnitude of the observation/action noise appears to be too small. With such minimal noise, there is not enough significant change to observe robustness, and the performance of all baselines seems to drop linearly. It might be more informative to increase the magnitude of the observation/action noise, as shown in Figure 7, to better demonstrate robustness.
2. The robustness experiment results regarding mass scale changes seem inconsistent between Figure 2 and Figure 7. This appears to be due to differences in the hyperparameter $I\_p$. If $I\_p$ is set to -1.0, as in Figure 7, the results might show improved robustness.
3. The experiments in non-periodic environments, such as those in metaworld, appear to have been conducted on too few environments. Expanding the evaluation to a broader range of environments could more convincingly demonstrate the effectiveness of MTC in non-periodic settings.
4. It would be beneficial to include a graph depicting behavior in non-periodic environments. Such a graph could help verify whether unnecessary state and action noise is effectively reduced in these cases as well.

**Methods And Evaluation Criteria:**

The total correlation maximization in MTC enables more efficient prediction of states and actions, and it is intuitively expected to work well for periodic tasks like those in DMC. However, in non-periodic tasks, there may be inherent limitations to the benefits of increasing total correlation. For instance, in Figure 4, while MTC shows superior performance compared to other baselines in the button-press-wall environment, the differences in other environments appear less significant. A more detailed explanation of the role and effects of total correlation maximization in non-periodic tasks would be appreciated.

**Other Comments Or Suggestions:**

1. Figure 2 is reported only in normalized scores; it would be preferable to also report episode rewards so that the results are consistent with the other evaluations.
2. Including the $I\_p$-related experiments from Figure 7 in the main paper could be beneficial, as the graph demonstrating that a stricter trajectory consistency constraint increases robustness serves as compelling evidence for the paper’s main claim.

**Other Strengths And Weaknesses:**

There are no additional strengths or weaknesses to mention.

**Questions For Authors:**

1. Could you also show the learning curve for Table 1? While the final performance is impressive, observing the stability and learning speed during training might provide more detailed insights into MTC’s performance.
2. Have you conducted experiments on other metaworld tasks with non-periodic environments (e.g., drawer-close, reach, etc.)? Presenting evaluations across a wider range of environments could further demonstrate the method’s high performance in non-periodic settings.
3. In Figure 4, does the right-hand graph include experimental results regarding robustness in non-periodic environments?

**Relation To Broader Scientific Literature:**

In the existing literature, algorithm performance and robustness are improved by removing unnecessary noise from either state or action sequences. This study also follows that research direction; however, while previous works have considered only the state sequence or the action sequence, this study takes into account the entire state-action trajectory. To achieve this, total correlation is introduced, and a lower bound is derived and utilized for the practical algorithm.

**Theoretical Claims:**

In Appendix A.1’s derivation of eq (2), it appears that eq (11) may be incorrect. Specifically, at L589, the terms $p(z\_{t+1})$ and $p(a\_t)$ should be defined as $\int\_{s\_{t+1}} p(z\_{t+1} | s\_{t+1} ) p(s\_{t+1}) ds\_{t+1}$ and $\int\_{s\_{t}} p(a\_{t} | s\_{t} ) p(s\_{t}) ds\_{t}$, respectively. Therefore, it does not seem appropriate to simply substitute them with $p(z\_{t+1} | s\_{t+1} )$ and $p(a\_{t} | s\_{t} )$ without including the integrals.

---

> ### Author Rebuttal · Authors · 2025-04-01
>
> Thank you for thoroughly reviewing our work and appreciating the performance and robustness of our method, and the persuasive evidence of our main claims.
>
> > benefit of total correlation maximization in non-periodic tasks
>
> Our main hypothesis that simple behavior that does not overfit to slight variations is less brittle is not restricted to periodic tasks, but also well-motivated for non-periodic tasks, such as manipulation tasks. For example, consider two different policies for grasping an object. The first policy is able to achieve high reward by always using essentially the same grasping motion and no adaptation. The second policy achieves similar reward, but relies more strongly on feedback and adapts even to slight variations. We expect the first behavior, which would have higher total correlation and better predictability but no periodicity, to be more robust to perturbations that have not been observed during training.
>
> > more non-periodic experiments
>
> We conducted additional experiments on five non-periodic MetaWorld tasks. On all tasks we achieved similar or better success rate than the baselines, supporting our hypothesis that regularizing towards simpler behavior is also beneficial for non-periodic tasks. A table with the results can be found in our reply to reviewer ts9N.
>
> > Derivation of eq (11)
>
> Our derivations do not assume that $\log(p(z\_t))$ is equal to $\log(p(z\_t|s\_t))$. However, due to the non-negativity of the expected KL $E\_{p(s\_t)} [ \text{KL}(p(z\_t|s\_t) || p(z_t))]$, substituting $p(z\_t|s\_t)$ for $p(z\_t)$ or $\pi(a\_t|s\_t)$ for $p(a\_t)$, does not invalidate our lower bound.
>
> > magnitude of noise
>
> Thank you for this constructive feedback. We use stronger state and action noises now. The following tables compare the performance (mean and 90% confidence interval over 20 seeds) on all eight tasks. Overall, MTC obtains better performance in the presence of strong noise.
>
> | Action noise with scale 0.5 | Acrobot Swingup | Hopper Stand | Finger Spin | Walker Walk | Cheetah Run | Quadruped Walk | Walker Run | Walker Stand |
> |:---------------------------:|:---------------:|:------------:|:-----------:|:-----------:|:-----------:|:--------------:|:----------:|:------------:|
> |             MTC             |    **159 ± 19**     |   **711 ± 36**   |  **643 ± 14**   |   **921 ± 9**   |   **637 ± 8**   |    **928 ± 8**     |  **396 ± 12**  |   **975 ± 3**    |
> |             RPC             |    111 ± 22     |  300 ± 115   |  **609 ± 24**   |  832 ± 54   |  539 ± 46   |    825 ± 80    |  **388 ± 25**  |   967 ± 4    |
> |           LZ-SAC            |     27 ± 7      |    19 ± 3    |  346 ± 26   |  715 ± 47   |  278 ± 12   |    430 ± 84    |  295 ± 9   |   905 ± 20   |
> |             SAC             |    108 ± 19     |  485 ± 132   |  **627 ± 27**   |  857 ± 30   |  502 ± 24   |   686 ± 111    |  356 ± 18  |   970 ± 6    |
>
> | Obs noise with scale 0.3 | Acrobot Swingup | Hopper Stand | Finger Spin  | Walker Walk |   Cheetah Run   | Quadruped Walk |  Walker Run  | Walker Stand |
> |:------------------------:|:---------------:|:------------:|:------------:|:-----------:|:---------------:|:--------------:|:------------:|:------------:|
> |           MTC            |    **8 ± 2**    | **190 ± 70** | **138 ± 12** |  **851 ± 10**   |      48±4       |  **901 ± 23**  | **351 ± 18** | **975 ± 2**  |
> |           RPC            |     **6 ± 1**     |   44 ± 30    |    28 ± 8    |  769 ± 35   |     54 ± 9      |    825 ± 82    | **341 ± 22** |   961 ± 8    |
> |          LZ-SAC          |      **3 ± 5**      |   **218 ± 22**   |   97 ± 11    |  755 ± 34   |   **101 ± 5**   |    567± 113    |   **334 ± 15**   |   955 ± 13   |
> |           SAC            |      **7 ± 1**      |   102 ± 44   |   104 ± 13   |  775 ± 29   |     48 ± 5      |   672 ± 110    |   **332 ± 26**   |   945 ± 21   |
>
> > Inconsistency between Figure 2 and Figure 7
>
> The inconsistency is caused by using different $I_p$, which we did not tune for each task in our robustness experiments.
>
> > depicting behavior in non-periodic environments
>
> As we are not able to show a graph in our reply, we evaluated the action-predictability to demonstrate that our regularizer also improves simplicity for non-periodic tasks. The table can be found in our reply to Reviewer 8GSd.
>
> > Related approaches using Fourier transform [1,2]
>
> Thank you for suggesting these methods, which are related to MTC and will be discussed in our revision. However, they are not closely related, since they do not consider the simplicity of behavior.
>
> > Normalized scores and learning curves.
>
> We can not show the learning curves here, but will add them to the revision with original rewards. MTC shows good learning stability.
>
> > Figure 7
>
> We will put Figure 7 in the main in our revision.
>
> > Figure 4
>
> No, Fig. 4 does not show  robustness in non-periodic tasks. We focused on DMC for our robustness experiments to follow the setup of closely related work (RPC and LZ-SAC).

---

> > ### Comment · Reviewer_vAa9 · 2025-04-04
> >
> > Thank you for the detailed response.
> >
> > I now understand the authors’ point that reducing unnecessary perturbations can be beneficial even in non-periodic environments. The results showing improved robustness in the presence of environmental noise support this claim.
> >
> > All of my concerns have been addressed, and I will update my score from 2 to 3.
> >
> > Lastly, regarding the predictability experiment table in the rebuttal to reviewer 8GSd—are the reported values negative log-likelihoods? If the prediction error is low, we would expect the log-likelihood to be high, or equivalently, the negative log-likelihood to be low. Could the authors clarify this?

---

> > > ### Author Response · Authors · 2025-04-07
> > >
> > > > are the reported values negative log-likelihoods?
> > >
> > > Sorry for the lack of clarity. The tables indeed show negated log-likelihood, indicating that the actions generated by our policies are more easily predicted.
> > >
> > > We sincerely appreciate that you raised your score and are now leaning toward accepting our work. Since this is our final opportunity to engage in this thread under this year’s ICML review process,
> > > we would like to take the opportunity to provide additional evidence that we could not fit in our last reply due to character constraints:
> > >
> > > 1. To demonstrate the learning stability of MTC, we now show the intermediate rewards (mean and 90% confidence interval over 20 seeds) on all 8 tasks every 200K steps.
> > >
> > > | Score | Acrobot Swingup | Hopper Stand | Finger Spin | Walker Walk | Cheetah Run | Quadruped Walk | Walker Run | Walker Stand |
> > > |:-----:|:---------------:|:------------:|:-----------:|:-----------:|:-----------:|:--------------:|:----------:|:------------:|
> > > | 200K  |     43 ± 10     |  481 ± 129   |  866 ± 21   |  897 ± 40   |  472 ± 47   |    294 ± 69    |  595 ± 28  |   959 ± 10   |
> > > | 400K  |     97 ± 17     |   792 ± 97   |  956 ± 12   |   959 ± 3   |  757 ± 28   |    724 ± 95    |  700 ± 21  |   976 ± 2    |
> > > | 600K  |    150 ± 22     |   867 ± 79   |   975 ± 9   |   962 ± 2   |  826 ± 26   |    792 ± 90    |  741 ± 17  |   972 ± 5    |
> > > | 800K  |    190 ± 21     |   923 ± 14   |   984 ± 2   |   962 ± 2   |   856 ±24   |    869 ± 60    |  774 ± 12  |   979 ± 5    |
> > >
> > > 2. We now evaluate the robustness to observation noise, action noise, and mass changes in all 8 manipulation tasks from Metaworld. Overall, MTC achieves better performance (mean and 90% confidence interval over 20 seeds) when actions
> > > and dynamics are perturbed, while being comparable to baselines in the presence of observation noise.
> > >
> > > | Action noise with scale 2.0 | Handle-pull-side-v2 | Drawer-open-v2  | Plate-slide-back-v2 | Peg-insert-side-v2 |    Sweep-v2     | Button-press-wall-v2 |  Door-lock-v2   |  Push-back-v2   |
> > > |:---------------------------:|:-------------------:|:---------------:|:-------------------:|:------------------:|:---------------:|:--------------------:|:---------------:|:---------------:|
> > > |             MTC             |   **0.98 ± 0.01**   | **0.60 ± 0.13** |   **0.71 ± 0.06**   |  **0.01 ± 0.01**   | **0.01 ± 0.01** |   **0.24 ± 0.04**    | **0.84 ± 0.03** | **0.10 ± 0.04** |
> > > |             RPC             |     0.72 ± 0.05     |   0.21 ± 0.13   |   **0.67 ± 0.08**   |    0.00 ± 0.00     |   0.00 ± 0.00   |   **0.26 ± 0.05**    |   0.69 ± 0.06   | **0.08 ± 0.03** |
> > > |             SAC             |     0.62 ± 0.12     |   0.19 ± 0.10   |     0.58 ± 0.07     |    0.00 ± 0.00     |   0.00 ± 0.00   |   **0.31 ± 0.06**    |   0.71 ± 0.06   |   0.01 ± 0.01   |
> > >
> > >
> > > | Mass change with scale 1.75 | Handle-pull-side-v2 | Drawer-open-v2  | Plate-slide-back-v2 | Peg-insert-side-v2 |    Sweep-v2     | Button-press-wall-v2 |  Door-lock-v2   |  Push-back-v2   |
> > > |:---------------------------:|:-------------------:|:---------------:|:-------------------:|:------------------:|:---------------:|:--------------------:|:---------------:|:---------------:|
> > > |             MTC             |   **0.98 ± 0.02**   | **0.62 ± 0.13** |   **0.98 ± 0.01**   |  **0.07 ± 0.06**   | **0.29 ± 0.10** |   **0.51 ± 0.08**    | **0.96 ± 0.02** | **0.18 ± 0.07** |
> > > |             RPC             |     0.71 ± 0.07     |   0.22 ± 0.16   |     0.87 ± 0.08     |    0.00 ± 0.00     |   **0.18 ± 0.09**   |   **0.59 ± 0.12**    |   0.84 ± 0.05   | **0.12 ± 0.04** |
> > > |             SAC             |     0.65 ± 0.12     |   0.21 ± 0.15   |     0.93 ± 0.03     |    0.00 ± 0.00     |   **0.18 ± 0.12**   |   **0.62 ± 0.10**    |   0.86 ± 0.05   |   0.00 ± 0.00   |
> > >
> > >
> > > | Obs noise with scale 0.05 | Handle-pull-side-v2 | Drawer-open-v2  | Plate-slide-back-v2 | Peg-insert-side-v2 |    Sweep-v2     | Button-press-wall-v2 |  Door-lock-v2   | Push-back-v2 |
> > > |:------------------------:|:-------------------:|:---------------:|:-------------------:|:------------------:|:---------------:|:--------------------:|:---------------:|:------------:|
> > > |           MTC            |   **0.95 ± 0.04**   | **0.45 ± 0.15** |     0.29 ± 0.09     |    **0.00 ± 0.00**     | **0.02 ± 0.01** |   **0.27 ± 0.04**    | **0.76 ± 0.07** | **0.00 ± 0.00**  |
> > > |           RPC            |     0.74 ± 0.07     |   0.18 ± 0.11   |     0.29 ± 0.08     |    **0.00 ± 0.00**     | **0.02 ± 0.02** |   **0.24 ± 0.08**    |   **0.74 ± 0.04**   | **0.00 ± 0.00**  |
> > > |           SAC            |     0.69 ± 0.13     |   0.13 ± 0.08   |   **0.46 ± 0.07**   |    **0.00 ± 0.00**     | **0.01 ± 0.01** |   **0.16 ± 0.07**    |   **0.73 ± 0.04**   | **0.00 ± 0.00**  |
> > >
> > >
> > > We hope this additional evidence fully convinces you of the significance of our work, and we kindly ask you to consider whether it might warrant an even higher score. Thank you again for your thoughtful feedback and engagement throughout the review process.

---

### Official Review · Reviewer_8GSd · 2025-03-13

**Overall Recommendation:** 4

**Summary:**

This paper proposes a method that learns compressible policies via a lower bound on the total correlation over a trajectory.
This is motivated by aiming to increase robustness by learning more simplistic policies.
In practice, the method trains a recurrent latent state and action predictor and adds the predictability of actions to the reward function.
In experiments, the authors show that their method leads to more robust and compressible policies.

**Claims And Evidence:**

* The main claim of the paper is that policies with a compressibility are more robust to disturbances, and that their method creates more compressible policies.

The improved robustness is well supported by experiments with distribution shifts.

 * They also claim that it improves robustness to spurious correlations.

This claim seems reasonable, but experiments do not directly evaluate it. The authors write that Gaussian noise is a strong distractor, but that's qualitatively different from an actual spurious correlation/ an actual distrator.


* The correlation or predictability of their learned policy is evaluated by compressing trajectories with an off-the-shelf lossless compression method and by visual inspection.

A more interpretable way to measure it might be to train predictors for each and evaluate their accuracies, or to actually evaluate the lower bound of total correlation derived by the authors

**Essential References Not Discussed:**

I am not aware of missing references.

**Experimental Designs Or Analyses:**

The  main experiments are sound.

Some experiments related to the evaluation of correlation could be done more directly and the evaluation of spurious correlation only with observation noise is insufficient.

**Methods And Evaluation Criteria:**

As stated above, the main experiments and evaluation are reasonable, but especially the claim of reducing spurious correlation should be investigated more directly or removed.

Overall it's reasonable.

Showing 90% CIs instead of the usual 95% is surprising, but it is clearly stated in the figure captions so that's fine.

**Other Comments Or Suggestions:**

L432 (right): "goal post" should be "goalpost

**Other Strengths And Weaknesses:**

The presentation/writing is very good.

**Questions For Authors:**

My main concern is the described non-markovianness of the reward as well as the non-stationarity of the rward due to the changing  predictor parameters.

How was this addressed in practice, or why is this not a problem? Maybe I am misunderstanding something.

If this can be addressed I am willing to raise my score.

**Relation To Broader Scientific Literature:**

The relation to prior method is discussed appropriately in the paper.

**Theoretical Claims:**

I checked the derivation of the proof for the lower bound and did not find any mistakes.

My main concern is that the reward formulation in (4) depends on all previous actions and states in an episode, as well as on the learned predictors.

This makes the reward formulation non-markovian and dependent on predictor parameters $\phi$ and $\theta$, and the notation as $r^*(s_t,a_t,s_{t+1})$ is questionable, it should be $r^*(s_{1:t},a_{1:t},\theta,\phi)$.

The paper claims that this can be "optimized straightforwardly with existing RL methods".

This is in no way obvious, it is not clear why we can simply ignore this dependency.

---

> ### Author Rebuttal · Authors · 2025-04-01
>
> Thank you for carefully reviewing our submission, and acknowledging the soundness of the main experiments, the good presentation and the empirical results that show more robust and compressible policies.
>
> > the claim of reducing spurious correlation should be investigated more directly or removed
>
> We evaluated the effect of spurious correlation in Appendix C4. We did not add Gaussian noise to the observation, but added additional state dimensions that are not controllable by the actor, but instead follow a fixed Gaussian transition model. These distractors are not correlated with the remaining states, the actions nor the reward that the agent receives, although by coincidence, it might appear that such correlations exists, resulting in spurious correlations.
> Our results show that MTC achieves higher rewards than RPC and SAC on the Walker Stand task, suggesting that MTC improves robustness to spurious correlations.
>
> > Some experiments related to the evaluation of correlation could be done more directly
>
> Thank you for suggesting evaluating the predictability to more directly assess the total correlation. We trained a t-step-ahead predictive model for t=[3,5,8,10] time steps on data sets that have been collected with the policy that has been learned with the different methods for a locomotion task and a manipulation task. The model was trained to predict the action t steps ahead, when given the current action. For all tested time differences, the prediction error of MTC was the smallest among all baselines.
>
> | test log-likelihood (Finger-spin) | t=3  | t=5 | t=8 | t=10 |
> |:--------------------------:|:---------------:|:---------------:|:---------------:|:-----------:|
> |       MTC        | **0.22** | **0.32** | **0.48**    |   **0.67**  |
> |       RPC        | 0.34  |  0.58  |  0.72   |  0.76   |
> |       LZ-SAC     |  0.96     |  1.31  |  1.48   | 1.53    |
> |       SAC        |  0.78   | 1.13   |   1.47  |   1.57  |
>
>
> | test log-likelihood (Drawer-open-v2) |    t=3    |    t=5    |    t=8    |   t=10    |
> |:--------------------------:|:---------:|:---------:|:---------:|:---------:|
> |       MTC        | **-4.84** | **-4.40** | **-4.05** | **-3.35** |
> |       RPC        |   -2.62   |   -3.05   |   0.14    |   0.17    |
> |       SAC        |   -2.37   |   -2.07   |   -2.7    |   -1.7    |
>
> > non-markovianness of the reward as well as the non-stationarity of the reward
>
> Although we clearly state that our augmented reward depends on the history-based decoder, we agree that the notation should reflect this dependency, and that the implication for the optimization should be discussed, and we will revise our submission accordingly.
> Our reward function depends on past states and actions to encourage the agent to act consistently with its previous actions. For our experiments, however, we did not provide the past states and actions to the policy, which limits its ability to perform optimally due to the non-Markovianity of its state-action-space. We did not provide the history to the policy solely to improve the simplicity of our method and the fairness of the comparisons. However, we performed an additional experiment on the Finger Spin task for the rebuttal to evaluate our approach in a Markovian setting (by using a recurrent policy), which did not result in significant differences.
>
> | Score (mean and 90% CI over 10 seeds) |     MTC     | MTC-History |
> |:-------------------------------------:|:-----------:|:-----------:|
> |              Performance              | **985 ± 2** | **986 ± 3** |
> |       Trajectores Size in Bytes       | **2993 ± 292** | **3031 ± 259** |
>
> We also agree that the non-stationarity of the reward should be discussed and we will revise the submission accordingly. Non-stationary rewards are quite common nowadays and are used effectively in different subfields, such as imitation learning, representation learning, and---most-relevant for us---by related works such as RPC and LZ-SAC. While establishing convergence guarantees for MTC (and the related works) would be desirable, we unfortunately have to leave this for future work.

---

> > ### Comment · Reviewer_8GSd · 2025-04-02
> >
> > I would like to thank the authors for their comprehensive reply.
> >
> > All my concerns were addressed, I'm raising my score (2->4)

---

> > > ### Author Response · Authors · 2025-04-02
> > >
> > > We greatly appreciate your approval of our work. Thank you!

---

### Official Review · Reviewer_moC5 · 2025-03-16

**Overall Recommendation:** 3

**Summary:**

An RL framework is presented that encourages total correlation within trajectories, leading to smoother, more periodic looking, and more compressible trajectories. This leads to robustness to observation noise, action noise, and robustness to changes in dynamics. The proposed algorithm is employed for DeepMind environments where it surpasses related work in expected reward, robustness, and compression.

**Claims And Evidence:**

Claims:
- MTC leads to robustness
- MTC leads to less "unnecessary variation", and more periodic trajectories
- MTC leads to compressibility

Good Evidence:
- Robustness evidence presented for one environment
	- Observation noise is added and MTC seems clearly more robust
	- Action noise is added and MTC seems clearly more robust
	- Dynamics are modified and MTC seems clearly more robust
- MTC generally better than other algorithms in DMC tasks

But Questions Arise
- is MTC better because of total correlation, or because of it being non-Markovian? While it is true that total correlation would induce a more periodic behaviour, it is harder for a Markovian policy to exhibit periodicity. We see this trend in this paper as well: SAC is Markovian, LZ-SAC uses a history of some states, RPC uses the complete history but only of states, and MTC uses the complete history of state-action pairs. And the the trend in performance is similar from worst to best: SAC, LZ-SAC, RPC, MTC. In other words my question is, is a good policy (regardless of total correlation) just more periodic and it will be learnt if it is non-Markovian, or does total correlation itself has a significant role to play.
- Robustness claims would be more solid if the other algorithms did not start worse off. A counter point can be made that since other algorithms start worse-off, addition of noise in actions/observations/dynamics causes degradation relative to how good a policy was originally.
- The main hypothesis: that a policy that exhibits more periodic behaviour is less brittle, I think needs better substantiation. It might be very true that some domains like such policies where is there is repetitive behaviour, but there might be others that don't . That does not take away anything from the core contribution of the paper, but the language I feel is a bit exaggerated if we don't have a strong theoretical proof that periodicity <-> robustness. (But I see that compressibility has been proposed as a good thing to have in other related works)

**Essential References Not Discussed:**

none that I am aware of

**Experimental Designs Or Analyses:**

Did not run any code. The design description in paper and appendix seems sound.

**Methods And Evaluation Criteria:**

Yes. Popular benchmarks. Very thorough experimentation. Comparison to relevant work.

**Other Comments Or Suggestions:**

I am not so sure about the use of the word "Coherence" for trajectories that show periodic behaviour or more total correlation. But maybe the authors are a better judge for that. But I highly suggest reviewing this term.

I don't understand p.5 col 2 last paragraph where it is said that rounding is necessary otherwise it leads to high variance in the results. Since rounding ignores the least significant bits, shouldn't it be the case that variance is not much affected by rounding?

**Other Strengths And Weaknesses:**

Strengths:
- Good, thorough experiments that empirical evidence of the claims in DMCS tasks.

**Questions For Authors:**

- Is periodicity -> robustness, or that periodicity in these domains = good policy and therefore high reward and more robustness?
- How much of this is due to non-Markovian nature, and how much due to explicit introduction of total correlation term?

**Relation To Broader Scientific Literature:**

A good contribution where we desire policies that exhibit more periodic behaviour. Good contribution in continuous control domains. Possibly the next natural step after RPC etc that the work is compared to.

**Theoretical Claims:**

Not in depth. No issues from a surface level reading except one: I don't understand why there is the term $r(s_T, a_T)$ in equation (3). i.e. the reward of last state-action pair ?

---

> ### Author Rebuttal · Authors · 2025-04-01
>
> Thank you for carefully reviewing our submission, and appreciating our contributions, and thorough experimentations.
>
> > is MTC better because of total correlation, or because of it being non-Markovian?
>
> While our total correlation objective results in a reward function that depends on the past in order to encourage the agent to perform actions that are consistent with its previous actions, we did not choose a history-based policy in our experiments for a simpler and fairer comparison. We ran additional experiments to test the effect of history-based observations for standard SAC and our method on the Finger Spin task. The results are consistent with non-history-based policies, indicating that the improvements are caused by our total correlation objective, whereas non-markovianity is not sufficient.
>
> | Score (mean and 90% CI over 10 seeds) |     MTC     | MTC-History |    SAC     | SAC-History |
> |:-------------------------------------:|:-----------:|:-----------:|:----------:|:-----------:|
> |              Performance              | **985 ± 2** | **986 ± 3** |955 ± 18 |  951 ± 23   |
> |       Trajectores Size in Bytes       | **2993 ± 292** | **3031 ± 259** | 5156 ± 526 | 5259 ± 442  |
>
> > Robustness claims would be more solid if the other algorithms did not start worse off
>
> Maintaining high performance in the face of noise or dynamics mismatch is significantly harder than maintaining low performance. Indeed, the performance of the worst possible policy could not degrade when subjecting it to observation noise. Instead, we argue that it is more relevant to consider the absolute performance in such test settings, rather than considering the changes relative to the training performance.
>
> > The main hypothesis, a policy that exhibits more periodic behavior is less brittle, needs better substantiation.
>
> We want to clarify that our primary interest is in learning ***simple*** behavior, which may, but does not need to manifest in periodic behavior. We challenged the hypothesis that simpler behavior tends to be less brittle on various periodic and non-periodic tasks, but did not encounter detrimental effects. We did not intend to claim that periodicity always increases robustness, and will carefully revise our submission to improve clarity. We will also revise the introduction to better motivate our hypothesis:
>
> During training, the reinforcement learning agent, for example a robot, typically observes many slight variations in the state, for example due to sensor noise or unmodelled dynamic effects. In some cases, it can be crucial for the agent to drastically adapt its behavior to react to these variations, but sometimes they can be safely ignored. We aim to bias the agent towards ignoring such variations, whenever doing so does not seriously decrease the expected return, and thereby follow Newton's first rule of his third book of principia: "We are to admit no more causes of natural things than such as are both true and sufficient to explain their appearances". Intuitively, we expect a given behavior that performed well for previous variations, to also perform well for future variations. We introduce this inductive bias by means of the additional objective of maximizing the total correlation within the trajectory produced by the agent. This total correlation corresponds to the amount of information that we can save by using a joint encoding of all (latent) states and actions within trajectories, compared to compressing all time steps independently. By maximizing total correlation, the agent is encouraged to produce compressible and predictable trajectories, and thereby biased towards open-loop behavior such as clean periodic gaits, without preventing it from performing adaptations when necessary.
>
> > why there is the term $r(s_T, a_T)$ in equation (3)?
>
> Thank you for spotting this mistake. The term belongs outside of the summation.
>
> > I am not so sure about the use of the word "Coherence" for trajectories that show periodic behaviour or more total correlation
>
> Thank you for this feedback. We used coherence mostly interchangeably with consistency, but will only refer to consistency in the revision.
>
> > [why rounding is necessary]
>
> From the point of lossless compression, the least significant bits are just as relevant as the most significant ones. When directly writing the floating point numbers of the simulator into a file, most of the bits in that file are used to encode digits that are beyond the precision of the control. When compressing such a file, the resulting file size will largely depend on how effectively those essentially random bits can be compressed, which significantly increases the variance of the resulting file size.
>
> > Is periodicity -> robustness?
>
> We do not believe that periodicity implies robustness and did not intend to make such a claim. We hypothesize that an inductive bias towards simpler behavior that avoids ***unnecessary*** adaptions can often improve robustness.

---

> > ### Comment · Reviewer_moC5 · 2025-04-07
> >
> > Thank you for these experiments and other experiments that I see in other rebuttals, and also for responding to our concerns with clarifications and explanations.
> >
> > I am increasing my score.
> >
> > Some questions still remain: In SAC-history are the rewards non-Markovian as given in MTC or just observations? I do not want to push for further experiments but this would further solidify the claims or give a direction for further investigation.
> >
> > About robustness, indeed you are right that a bad policy will not be made worse by further addition of noise and it is easier to maintain the level of a worse policy than a better policy, but I am hesitant to fully accept the claim in fig 2-left for example. Maybe we can imagine these to be comparable in the small region where all are performing well) An increase in noise strength of 0.02 results in average reward drop of 0.2. But then again there is no reason there should be a linear relationship between noise strength and reward, even in this region. But a case of robustness is more strongly made when one method drops off to near-random / random quickly (e.g. in fig2-center) but the robust one does not.

---

> > > ### Author Response · Authors · 2025-04-08
> > >
> > > > [Rewards for SAC-History]
> > >
> > > We indeed only used the original rewards for SAC-history because we did not understand which non-Markovian reward we could provide. We now understand that you are suggesting to use the MTC reward, which will lead to an ablation of MTC that does not backpropagate the lower bound directly into policy and encoder, but only uses its effects on the reward to come. We agree that such ablation would be interesting, in particular as the resulting algorithm would be better decoupled from the underlying RL algorithms.
> > >
> > > However, we argue that such experiment would not be suitable to test whether the improvements are caused by our total correlation objective, since the MTC-reward is derived based on total correlation maximization after all. In MTC the total correlation regularizer has two effects: 1) modifying the reward function and 2) directly modifying the gradient to policy & encoder during the policy improvement step. So far, we only demonstrated that both effects in total positively impact the performance, robustness, compressibility and predicatbility of the resulting behavior. As our lower bound with history-based prediction models causes the desired effects, we believe that it would also be beneficial when only using it for 1) or 2). But, of course, we would need to investigate this in future work.
> > >
> > > For the current submission, we focused on the algorithm that is consistent with our derivations. We argue that an algorithm that is derived from a principled objective is desirable by providing more insights into the underlying mechanics (e.g. biasing towards history-based predictions improves total correlation) and may also lead to more stable algorithms (updating all models with respect to the same bounded objective ensures convergence under the assumption---which admittely is strong in the RL setting---that this objective is indeed improved during every update). However, we agree that only using the lower bound for computing the reward could perform similarly well in practice, which would be of large interest for the community, for example, by providing a simple modification to PPO (which is almost exclusively used for real-robot locomotion) that can induce simpler, more robust behavior, without manual reward engineering.
> > >
> > > > there is no reason there should be a linear relationship between noise strength and reward
> > >
> > > We agree that there is no reason to expect the relationship between noise strength and reward to be linear. A constant increase in noise could result in moderate effects relative to the zero-noise level, larger effects when applied on top of some existing noise (due to compounding effects), and in negligible effects when applied to very large noise level where the policy is not able to perform reasonably anyway. Hence, it is difficult to compare the robustness to noise when the different policies already start at different reward levels. However, we noticed that MTC, RPC and SAC perform very similarly on Walker-Stand and investigated the robustness while focusing on this single environment. As shown below, robustness increases with respect to observation noise, action noise as well as mass change.
> > >
> > > | Obs noise    |  Scale=0.0  |  Scale=0.1  |  Scale=0.2  |  Scale=0.3   |  Scale=0.4   |  Scale=0.5   |
> > > |:------------:|:-----------:|:-----------:|:-----------:|:------------:|:------------:|:------------:|
> > > |     MTC      | **983 ± 2** | **984 ± 1** | **980 ± 2** | **975 ± 2**  | **960 ± 7**  | **924 ± 12** |
> > > |     RPC      | **980 ± 5** | **981 ± 2** | **978 ± 3** |   961 ± 8  |   911 ± 18 | 829 ± 24 |
> > > |    LZ-SAC    |   977 ± 2   |   976 ± 2   |   972 ± 3   |   956 ± 13 |   898 ± 24 | 779 ± 35 |
> > > |     SAC      | **985 ± 2** | **983 ± 1** | **975 ± 8** | 945 ± 21 | 895 ± 29 | 799 ± 33 |
> > >
> > > | Action noise |  Scale=0.0  |  Scale=0.2  |  Scale=0.4  |  Scale=0.6   |  Scale=0.8   |  Scale=1.0   |
> > > |:------------:|:-----------:|:-----------:|:-----------:|:------------:|:------------:|:------------:|
> > > |     MTC      | **983 ± 2** | **982 ± 1** | **977 ± 2** | **967 ± 5**  | **893 ± 15** | **718 ± 21** |
> > > |     RPC      | **980 ± 5** | **980 ± 2** | **976 ± 3** | **960 ± 4**  |   850 ± 15   |   661 ± 12   |
> > > |    LZ-SAC    |  977 ± 2    |   975 ± 2   |   961 ± 8   |   790 ± 25   |   588 ± 19   |   459 ± 15   |
> > > |     SAC      | **985 ± 2** | **981 ± 2** | **976 ± 4** | **949 ± 15** |   842 ± 22   |   666 ± 16   |
> > >
> > > | Mass change |  Scale=0.1  |  Scale=0.5  |  Scale=1.0  |  Scale=1.5   |  Scale=2.0   |
> > > |:-----------:|:-----------:|:-----------:|:-----------:|:------------:|:------------:|
> > > |     MTC     | **935 ± 6** | **982 ± 1** | **983 ± 2** | **962 ± 10** | **767 ± 34** |
> > > |     RPC     |  850 ± 24   | **978 ± 3** | **980 ± 5** | **963 ± 6**  | **738 ± 32** |
> > > |   LZ-SAC    |  706 ± 28   |   970 ± 3   |   977 ± 2   |   938 ± 20   | **708 ± 52** |
> > > |     SAC     |  858 ± 28   | **979 ± 2** | **985 ± 2** | **960 ± 7**  | **759 ± 33** |
> > >
> > >
> > > Thank you for your thoughtful feedback and engagement!

---

### Official Review · Reviewer_ts9N · 2025-03-19

**Overall Recommendation:** 2

**Summary:**

The paper proposes an extension to standard reinforcement learning by introducing a regularization objective that maximizes the total correlation across latent state representations and actions in an agent’s trajectory. It derives a variational lower bound on this total correlation, which is incorporated into the soft actor-critic framework to jointly optimize the policy, state encoder, and prediction models. Experimental evaluations on simulated robotic control tasks show that this approach yields more coherent, periodic, and compressible trajectories, leading to improved performance and enhanced robustness against observation noise, action perturbations, and dynamics changes. Overall, the paper argues that embedding total correlation as an inductive bias can foster simpler and more robust behaviors in reinforcement learning agents.

**Claims And Evidence:**

Most of the claims are supported by the experiments in simulation, which show improvements in consistency, robustness, and performance. However, one issue is that the method uses a lower bound for total correlation that is always negative, making it unclear how accurately it reflects the true value. Additionally, since the evidence comes only from simulated environments, it's not fully clear if the same benefits would apply in real-world situations.

**Essential References Not Discussed:**

Algorithms such as TD-MPC use BYOL losses to learn latent representations of the environment. Comparison to this literature would be relevant.

**Experimental Designs Or Analyses:**

Experiments are insufficient, as claims are entirely supported by DMControl experiments. Author is encouraged to apply their approach to a wider variety of environments.

**Methods And Evaluation Criteria:**

Most claims are based on DMControl results. Results should be evaluated across a wider range of benchmarks.

**Other Comments Or Suggestions:**

None for now.

**Other Strengths And Weaknesses:**

Strengths:
- Introduces a creative method that combines total correlation with reinforcement learning to improve policy robustness and consistency.

Weaknesses:
- Results limited to DMC.

**Questions For Authors:**

None for now.

**Relation To Broader Scientific Literature:**

Method attempts to imbue priors in existing RL algorithms to encourage temporal consistency and periodicity of movements. However, optimal policies in DMC environments exhibit periodic behavior regardless of additional priors. Unclear how total correlation affects this.

**Theoretical Claims:**

Did not check.

---

> ### Author Rebuttal · Authors · 2025-04-01
>
> Thank you for providing valuable comments and acknowledging the novelty and empirical performance of our approach.
>
> > Method uses a lower bound for total correlation that is always negative, making it unclear how accurately it reflects the true value.
>
> As stated under limitations, our lower bound is not meant for approximating the total correlation, but still effective for optimization, as demonstrated in our experiments that show simpler, more compressible, and more robust behavior. To further investigate the ability of our method to maximize the total correlation, we conducted additional experiments on the Finger spin task for this rebuttal. Namely, we collected data sets using the policies that have been learned with the different methods, and trained a model to predict the action t steps ahead, based on the current action. As indicated in the following table, our lower bound object significantly increases the predictability, further suggesting that MTC is effective in increasing the consistency, predictability and thereby total correlation of the emerging behavior.
>
> | test log-likelihood | t=3  | t=5 | t=8 | t=10 |
> |:--------------------------:|:---------------:|:---------------:|:---------------:|:-----------:|
> |       MTC        | **0.22** | **0.32** | **0.48**    |   **0.67**  |
> |       RPC        | 0.34  |  0.58  |  0.72   |  0.76   |
> |       LZ-SAC     |  0.96     |  1.31  |  1.48   | 1.53    |
> |       SAC        |  0.78   | 1.13   |   1.47  |   1.57  |
>
>
> > Not fully clear if the same benefits would apply in real-world situations.
>
> Our experiments show that policies that are trained with our regularizer exhibit a simpler behavior that is more robust to mismatches in the system dynamics. We argue that these properties are highly desirable when attempting to transfer these policies to a real robot. However, we unfortunately were not able to perform experiments on a real robot.
>
> > Author is encouraged to apply their approach to a wider variety of environments.
>
> We now additionally evaluated our approach on 5 more tasks from Metaworld. Also on these non-periodic
> manipulation tasks, MTC outperforms SAC and RPC in terms of average success rate, demonstrating the benefits of maximizing the total correlation
> on solving manipulation tasks. All in all, our experiments are quite diverse, span periodic and nonperiodic, vision-based and non-vision-based tasks, closely follow the testbeds of the most related works, and show strong performance in all of these settings.
>
> | Success rate at 500K steps (mean and 90% CI over 10 seeds) | Handle-pull-side-v2 | Drawer-open-v2  | Plate-slide-back-v2 | Peg-insert-side-v2 |    Sweep-v2     |
> |:----------------------------------------------------------:|:-------------------:|:---------------:|:-------------------:|:------------------:|:---------------:|
> |                            MTC                             |   **0.98 ± 0.02**   | **0.68 ± 0.18** |   **0.99 ± 0.02**   |  **0.06 ± 0.05**   | **0.28 ± 0.14** |
> |                            RPC                             |     0.79 ± 0.10     |   0.23 ± 0.21   |   **0.98 ± 0.03**   |    0.00 ± 0.00     | **0.23 ± 0.14** |
> |                            SAC                             |     0.76 ± 0.19     |   0.22 ± 0.21   |   **0.93 ± 0.07**   |    0.00 ± 0.00     | **0.19 ± 0.17** |
>
> > However, optimal policies in DMC environments exhibit periodic behavior regardless of additional priors. Unclear how total correlation affects this.
>
> While DMC tasks are characterized by periodic motions, our results in Fig.1, Fig.3, Fig.9 and Fig.10 clearly show that
> maximizing the total correlation improves the consistency and periodicity of behaviors compared to leading baselines. Furthermore, we are primarily interested in learning ***simple*** behavior, which does not necessarily imply periodicity, for example, for manipulation tasks.
>
>
> > [Comparison to literature such as TD-MPC and BYOL]
>
> We now provide the following discussion of these two approaches in our Related Work Section: Our approach is also related to
> previous approaches that extract temporally consistent representations from observations by learning latent dynamics models, such as TD-MPC and BYOL. Different from these approaches only considering temporal consistency in representations of states, our total correlation objective aims to maximize the consistency
> among the whole trajectories of state representations and actions. As shown in our experiments (Fig.5), additionally enforcing the consistency within action sequences by learning the action prediction
> model improves the performance in the presence of environmental perturbations.

---

### Decision · Program_Chairs · 2025-05-01

**Decision:**

Accept (poster)

**Comment:**

This paper adds a regularization to SAC to maximize the total correlation between action trajectories and latent representations. The objective is meant to yield a simpler / more consistent policy. The method shows robustness to various types of noise, and the experimental results appear to be consistent with the central motivations and claims.

There were some concerns about the claims in the paper wrt periodicity of the policy, tested environments, Markovian behaviors, etc. Some of these appear to have been addressed sufficiently, some still are somewhat unresolved. I have my own questions as well: the total correlation objective in Equation 1, as it includes the policy, can encourage the policy to move towards trivial / easy-to-predict states. This was a problem in earlier literature that attempted to use mutual information as a regularizer, and a careful balance was needed to ensure there was sufficient signal from the reward to avoid this sort of collapse. But the bound used here only has the policy parameters in the marginal terms (denominator, the joint action terms use an estimator q which doesn't appear to effect the policy parameters), which appears to mitigate this problem. I feel like this was an important point missed in the paper as well as the discussion, as this represents a discrepancy between the original objective and the final algorithm that non-trivially rectifies a known problem with using TC regularization on policies.

Despite this, for the most part the paper appears to be a good enough contribution. If possible I'd like to see some sort of discussion about the above, but I'm inclined to accept the paper.